# Compact and wideband nanoacoustic pass-band filters for future 5G and 6G cellular radios

Gabriel Giribaldi [1,2] ✉, Luca Colombo[1,2], Pietro Simeoni[1] & Matteo Rinaldi[1] ✉

Over recent years, the surge in mobile communication has deepened global connectivity. With escalating demands for faster data rates, the push for higher carrier frequencies intensifies. The 7–20 GHz range, located between the 5G sub-6 GHz and the mm-wave spectra, provides an excellent trade-off between network capacity and coverage, and constitutes a yet-to-be-explored range for 5G and 6G applications. This work proposes a technological platform able to deliver CMOS-compatible, on-chip multi-frequency, low-loss, wide-band, and compact filters for cellular radios operating in this range by leveraging the micro-to-nano scaling of acoustic electromechanical resonators. The results showcase the first-ever demonstrated low insertion loss bank of 7 nanoacoustic passband filters in the X-band. Most of the filters showcase fractional bandwidths above 3% and sub-dB loss per stage in an extremely compact form factor, enabling the manufacturing of filters and duplexers for the next generation of mobile handsets operating in the X-band and beyond.

Over the past 30 years, mobile communication has experienced unparalleled expansion in number of users, data transmission volume, and research and development efforts[1]. This technological revolution has paved the way to the remarkable level of connectivity that we enjoy in our everyday lives. The first ever SMS was sent as recently as 1992, followed by the first downloadable content sold to phones in 1998, while the mobile internet enabled by 3G started to be adopted worldwide in 2003[2].

Nowadays, the state-of-the-art features available to the consumer include social networking, live video-streaming, Internet of Things (IoT) nodes, Virtual Reality (VR), and more. These technologies rely heavily on both software and hardware solutions, and are taking advantage of the latest 5G roll-outs, allowing for record-breaking data-rates, spectral efficiency, and latency[3].

Nevertheless, the 5G mm-wave range is far from being optimally exploited, given the lack of high-performance commercial hardware solutions for the Radiofrequency Front-Ends (RFFEs) of handheld devices. Moreover, the high propagation losses of carrier waves at mm-wave frequencies ($f > 24$ GHz) demand for the presence of a large,

capillary network of base stations, notably increasing the cost of the physical infrastructure required to deploy the technology.

The 7–20 GHz range, on the other hand, provides an excellent trade-off between spectral efficiency and coverage, and is now being proposed as third Frequency Range (FR-3) for 5G. Moreover, it is foreseen to be exploited also in 6G to constitute its mid-band, devoted to crowded urban areas[4]. In this framework, the component carrier bandwidth is foreseen to be increased from 5G 100 MHz to 400 MHz or more[4]. These strict bandwidth requirements, combined with the high frequency of operation, directly translate into the need for novel and improved passive RFFE hardware for the band-pass filtering, with the fundamental tasks of improving the received Signal to Noise Ratio (SNR) and duplexing the transmitted and received signals.

As for the 5G mm-waves, the choice for high-performance filters that would fit in mobile handsets is scarce. Devices adopting electromagnetic (EM) resonances such as cavity and waveguide filters provide good performance, with low insertion loss (IL) and bandwidth (BW) of several GHz around 30 GHz[5], but the large footprint that characterizes these technologies makes them unfeasible to

[1]NanoSI Institute, Northeastern University, Boston, MA, USA. [2]These authors contributed equally: Gabriel Giribaldi, Luca Colombo.
✉e-mail: giribaldi.g@northeastern.edu; m.rinaldi@northeastern.edu

become the commercial standard. Microstrip filters, on the other hand, being constituted by thin metal strips on printed circuit boards (PCBs), can be integrated and are already part of commercial products for filtering in the mm-wave range[6]. Nevertheless, they suffer from wide manufacturing variability, temperature variations, and low performance in terms of bands roll-off. Therefore, they will be unable to provide high channel selectivity once the spectrum becomes more densely exploited.

Up to the current 5G sub-6 GHz band, state-of-the-art RFFE passive band-pass filters are being implemented with piezoelectric micro-electromechanical (MEM) resonators combining high quality factors ($Q$) and compact footprints[7]. These devices present a mechanical resonance at a specific frequency (namely the resonance frequency, $f_s$) depending on their geometry, confining the acoustic standing wave in the device's body with minimal loss between its terminals.

The main commercial piezoelectric RF MEMS technologies rely on Surface Acoustic Waves (SAWs)[8] and Bulk Acoustic Wave (BAWs) resonators[9,10] (Table 1). In SAW devices, the acoustic standing wave is ideally confined on the surface of the piezoelectric layer, while BAW devices experience mechanical stress across the whole resonator body. SAW technologies employ Interdigitated Electrodes or Transducers (IDTs) to excite the resonance, and thus allow to lithographically define the resonance frequency by varying the pitch of the IDTs. This provides design flexibility and cost-effectiveness allowing the fabrication of devices with different operating frequencies on the same substrate. Moreover, SAW resonators rely on an extremely simplified fabrication process. Recent commercial implementations of this technology include Temperature Compensated (TC)[11] and Thin-Film (TF) SAWs[12,13]. The main drawbacks of this technology are the high losses experienced at high frequency and the relatively low electromechanical coupling ($k_t^2$) of the resonant mode. Devices at higher frequencies have been demonstrated, but their performance or fabrication cost were found not suitable for mass commercial products[14,15]. The second drawback is generally circumvented by adopting single crystal materials such as lithium niobate (LN) and lithium tantalate (LT), also in the form of thin-films on insulators, allowing the harvesting of larger $k_t^2$ values thanks to their superior piezoelectric coefficients. However, the fabrication of these thin films is relatively expensive and hardly compatible with monolithic CMOS processing, making them unsuitable for direct integration with the RFFEs circuitry. In fact, MEMS-CMOS monolithic integration, even though not currently exploited in commercial filters, will prove to be indispensable when further increasing the operating frequency of RFFEs. In fact, it would allow not only a higher degree of miniaturization, but also a tighter control of non-idealities such as interconnect-related parasitic capacitances and inductances, leading to significant performance enhancement[16].

BAWs' market is dominated by thin-Film Bulk Acoustic Resonators (FBARs)[17], employing Aluminum Nitride (AlN) or lightly-doped (LD) Scandium AlN (ScAlN) as piezoelectric layer. These materials are cheaper than single-crystal thin-films, are compatible with CMOS Back-End Of the Line (BEOL) processes[18], and are currently used in state-of-the-art resonators for filtering applications. Monolithic integration of AlN resonators for filters, oscillators, fingerprint sensors, and energy harvesting applications has been successfully demonstrated[18,19]. Despite the lack of available literature, post-CMOS compatibility of ScAlN is widely accepted within the community, due to the similarity in deposition conditions and machining between AlN and ScAlN[20–22].

FBARs combine high $Q$ and $k_t^2$, allowing the synthesis of highly selective and low $IL$ band-pass filters[23]. Futhermore, FBAR overtones reaching 33 GHz[24] and microacoustic filters above 20 GHz[25] have been demonstrated. Moreover, they can count on a standardized large-scale fabrication process[26]. Along with FBARs, successful commercial devices exploiting bulk acoustic resonant modes include XBAW[27] and Solidly Mounted Resonators (SMRs)[28].

**Table 1 | Comparison of the main commercial RF MEMS resonator technologies and the LFE CLMR presented in this work**

| | Trad. SAW[8] | TC-SAW[11] | TF-SAW[12,13] | FBAR[17] | SMR[28] | XBAW[27] | CLMR (This work) |
|---|---|---|---|---|---|---|---|
| Substrate | LN/LT | LN/LT | LN/LT | AlN/ScAlN | AlN/ScAlN | AlN/ScAlN | AlN/ScAlN |
| CMOS compatibility | No | No | No | Yes | Yes | Yes | Yes |
| Litho tunability | Yes | Yes | Yes | No | No | No | Yes |
| Cost | Low | Medium | High | High | Medium | High | Low |
| Fab. complexity | Low | Low | Low | High | High | High | Low |
| Max. freq. | < 4 GHz | < 4 GHz | < 4 GHz | < 6 GHz commercially | < 6 GHz | < 6 GHz | >10 GHz |

Although FBAR (column 5) and XBAW (column 7) share similar structure, they differ significantly in fabrication process and associated challenges.

Since the FBARs' advent, there has been significant effort to overcome their intrinsic limitations. In fact, given their purely vertical excitation, the operating frequency depends on the piezoelectric film's thickness, requiring multiple mass-loading and trimming operations to shift their resonance and achieve on-chip frequency diversity. This procedure requires additional lithographic steps and tighter fabrication requirements which contribute to complexity and cost increase.

Additionally, the scalability of such devices to the high frequencies in the mm-wave and 7–20 GHz spectra is jeopardized by their lateral and vertical dimension shrinking, which magnifies the fabrication challenges associated to both frequency trimming and frame design for effective acoustic wave confinement[10].

For these reasons, devices whose resonance frequency can be lithographically defined have been intensely investigated with the goal of attaining performance levels on par with current BAW resonators while significantly simplifying their fabrication process[29–32]. These devices typically belong to the class of Lamb Wave Resonators (LWR), and they have historically been implemented on both LN and AlN platforms. LN-based devices, while demonstrating record-breaking FoM at high frequency[33–35], inherit the same CMOS integration issues and the high-costs of LN SAW resonators. Furthermore, recently developed periodically poled films (P3F) only ensure $k_t^2$ maximization at a single frequency, due to the strong relationship between the desired mode of vibration and the material stacking necessary to sustain it. This peculiarity prevents the manufacturing of devices with high performance capable of spanning the whole frequency band of interest. On the other hand, AlN is post-CMOS compatible, but its adoption for lithographically frequency-defined resonators has been limited because of its low $d_{31}$ piezoelectric coefficient, which leads to modest $k_t^2$ values when compared to FBARs[29].

A possible solution to this problem is represented by Cross-sectional Lamé Mode Resonators (CLMRs), which exploit a bi-dimensional mode of vibration. By hybridizing the $S_0$ and the $S_1$ modes, which are respectively excited via the $d_{31}$ and $d_{33}$ coefficient, $k_t^2$ as high as 7% have been recently demonstrated in an AlN thin-films sandwiched between a double-IDTs topology[36]. Nevertheless, this configuration is impractical at high frequency (>4 GHz) due to challenges tied to the growth of sputtered piezoelectric material on a patterned substrate. However, CLMRs can also be designed in a Lateral Field-Excited (LFE) configuration (Fig. 1a–c), removing the need for a bottom electrode and considerably simplifying the micro-machining process. Devices operating in the fundamental mode at 11 GHz[37] and overtones at 23 GHz[38] demonstrated quality factors as high as 600 and 250, respectively. Nevertheless, their $k_t^2$ performance (<1.3%) does not allow the synthesis of wide-band filters.

In recent years, Scandium doping has been proposed to increase AlN piezoelectric coefficients while enhancing material compliance[39,40]. This innovation led to the demonstration of lightly-doped (LD) ScAlN resonators with unprecedented $k_t^2$[41–44], which were promptly introduced into the market. However, due to the non-linear dependency between material properties and Scandium concentration, it is projected to attain maximum $k_t^2$ enhancement at high doping levels. Highly-doped (HD) ScAlN (Sc$_{[\%]}$≥30%) is therefore a promising material for LFE CLMRs, allowing to combine the advantage of simple fabrication and on-chip lithographic definition of the resonance frequency with $k_t^2$ values larger than AlN FBARs. In[43], preliminary results on resonators operating in X-band have been reported. However, the combination of non-optimal film crystallinity, surface roughness, and defect density on the material side, along with sub-optimal design at the device level, led to performance unsuitable for any commercial application. More recently, significant enhancement of the resonators' performance have been reported[45], thanks to thin film crystallinity improvements.

In this work, by leveraging the deposition of high-quality HD ScAlN thin-films, the first bank of X-band passive ladder filters is demonstrated. The devices span over 3 GHz of frequency, feature sub-dB insertion loss per ladder stage, and wide filter *BW*. Therefore, the CLMR technology is envisioned by the authors as a promising candidate to satisfy the need of compact and wide-band filtering devices in the 7–20 GHz range. The current work proposes an implementation of standalone CLMRs and CLMR-based filters with nanometric features (nanoacoustic devices) and unprecedented performance in the 6 to 20 GHz range, paving the way for more complex designs and experimentally demonstrating the outstanding capabilities of this technological platform.

## Results

### Highly-doped ScAlN

Scandium-doped AlN provides a considerable enhancement of the $d_{31}$ and $d_{33}$ piezoelectric coefficients, an increased dielectric constant ($\epsilon_r$), and a reduction of the material's stiffness. The evolution of the aforementioned properties is well captured by ab-initio equations derived from the Density-Functional Theory (DFT)[40], which exhibit a superlinear dependency with scandium concentration. Consequently, the most significant enhancement in $k_t^2$ occurs at higher doping levels (Sc$_{[\%]}$≥30%). Larger electromechanical couplings are obtained due to the combined increase of the piezoelectric coefficients and the softening of the equivalent stiffness constant[7]. As shown by a rapidly increasing volume of publications on the subject[46–48], the growth of highly c-axis oriented HD ScAlN is not trivial given the local lattice deformation introduced by the Sc atoms, which promote the growth of Abnormally-Oriented Grains (AOGs)[49], well-known to lower both $Q$ and $k_t^2$ of microacoustic devices[50]. Furthermore, AOG challenges worsen with increased film thickness and doping.

An industrial-grade Evatec® Clusterline 200 (Fig. 1d) in conjunction with a 12" Al$_{0.7}$Sc$_{0.3}$ alloy casted target was adopted to deposit high-quality ScAlN thin films. The deposition was carried out with reactive magnetron sputtering directly on 200 mm high-resisitivity Si < 100 > wafers, while targeting a film thickness of 280 nm. Good thickness uniformity was obtained across the whole wafers, as reported in Fig. 1e. More details on the sputtering recipe are provided in the Methods section. The film's crystallinity was assessed via X-Ray Diffractometry (XRD), showing a low Full Width Half Maximum (FWHM) value of 2.1˚ (Fig. 1f). To the authors' knowledge, this is the lowest value ever reported on a < 300 nm 30% ScAlN film sputter-deposited directly on silicon without a bottom electrode or an AlN seed layer. Moreover, the fabricated ScAlN layer only shows a slight FWHM degradation when compared to AlN films with similar thickness deposited with the same tool, while fall short when benchmarked against epitaxial films reported in literature[42]. Nevertheless, the sputtering process achieves lower film stress, higher throughput, and lower manufacturing cost, and it is therefore preferable for mass production. Moreover, resonators fabricated with the present sputtered film notably exceed the performance of similar devices built on epi-films, see below.

Figure 1 g–h shows two Atomic Force Microscope (AFM) images of the top surface of the film. More specifically, Fig. 1g depicts the center of the wafer, while Fig. 1h focuses on the edge, demonstrating that the film presents a smooth surface and low AOG density across the whole diameter while preserving sub-nm surface roughness. Finally, in Fig. 1i a Transmission Electron Microscope (TEM) image of the film's cross-section is reported, showing the material stack and good degree of c-axis orientation, especially considering the low film thickness.

### Fabrication and testing setup

The devices were fabricated on 200 mm Si wafers with standard micro-machining techniques and a three-mask process. The main steps are shown in Fig. 2a–b, while a detailed description of the fabrication process is reported in the Methods section. SEM pictures of the fabricated devices (both standalone resonators and nanoacoustic filters) are shown in Fig. 2c–f.

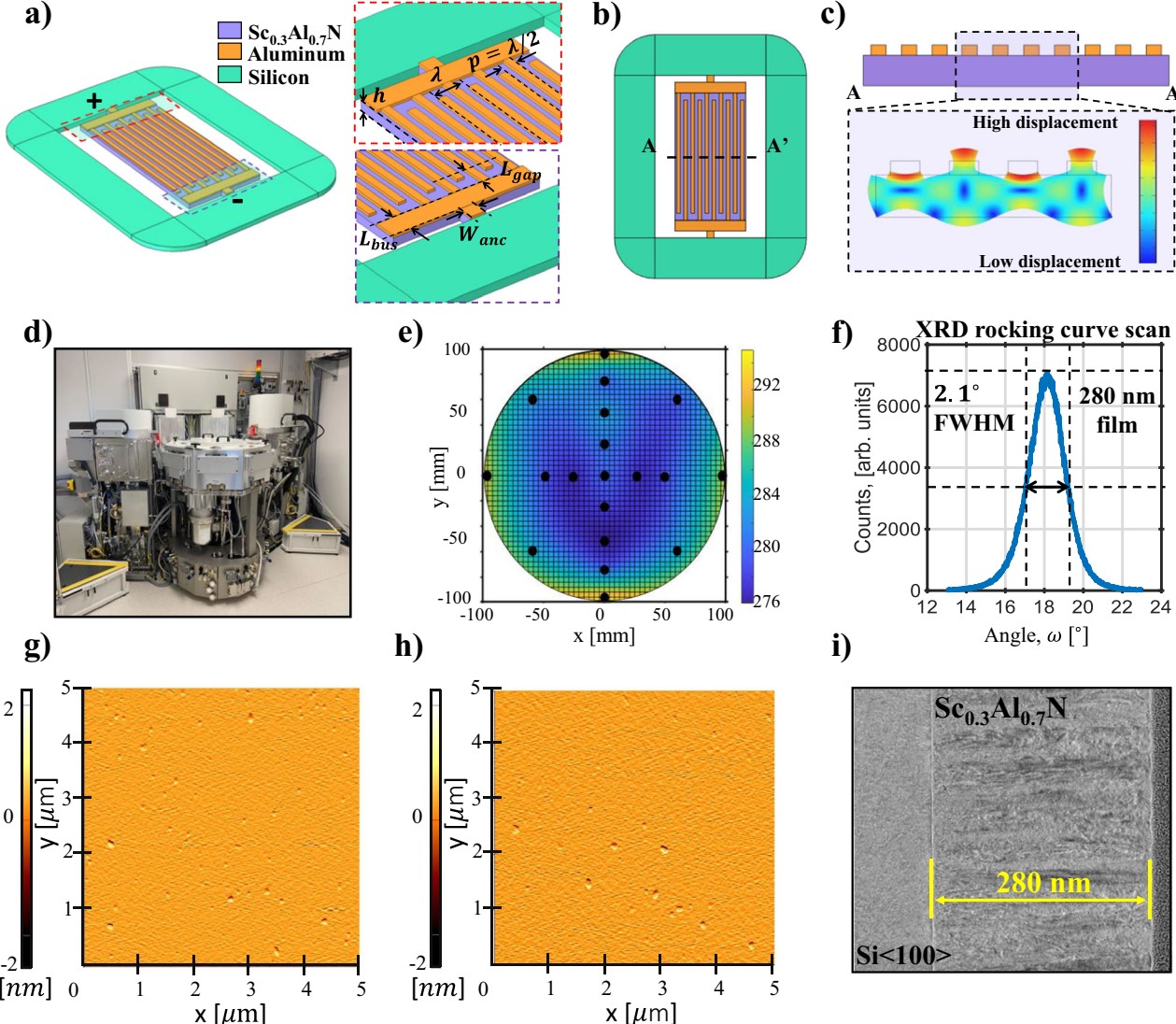

**Fig. 1 | Lateral field-excited CLMRs and ScAlN optimization. a** Cross-sectional Lamé Mode Resonator (CLMR) 3D view, with zoom-ins showing some of the most important geometrical dimensions, i.e. the piezoelectric film's thickness ($h$), the horizontal acoustic wavelength ($\lambda$), the pitch ($p = \lambda/2$), the bus length ($L_{bus}$), the anchor width ($W_{anc}$), and the gap length ($L_{gap}$). The different materials employed in this work are shown. In **b**, the top view of a CLMR is shown, while **c** shows its cross sectional view, along with the COMSOL® simulated mode shape at resonance. **d** Industrial-grade Evatec® Clusterline 200 sputtering tool employed to deposit the ScAlN thin films of this work. In **e**, a thickness map of the deposited film is shown. The black dots represents the measurements, which were later fitted with a 3D spline to create a thickness distribution heatmap. The measurements were performed via ellipsometry. In **f**, the X-Ray Diffractometry (XRD) rocking curve scan of the $Sc_{0.3}Al_{0.7}N$ 280 nm film with a measured FWHM value of 2.1° is reported. **g**, **h** show Atomic Force Microscope (AFM) top views of the film taken in two different locations, demonstrating a low density of AOGs and sub-nm surface roughness of 0.65 and 0.61 nm, respectively. **g** represents an AFM scan of the wafer's center, while **h** the AFM of an edge. In **i**, a Transmission Electron Microscope (TEM) cross-sectional view of the same film is reported, showing high c-axis orientation.

The devices were characterized in laboratory conditions in a two-port configuration by means of a Vector Network Analyzer (Keysight N5221A), two low-loss coaxial RF cables, a probe station, and two Cascade Ground-Signal-Ground (GSG) probes with 150 µm pitch or a Ground-Signal (GS) and a Signal-Ground (SG) probes for resonators and filters, respectively. The VNA was calibrated via Transmission-Reflection-Line (TRL) calibration kit standards.

**Standalone nanoacoustic resonators**

Leveraging CLMRs' outstanding lithographic definition of the resonance frequency, devices in the 6–11 GHz range of operation were designed and fabricated on the same chip (Fig. 3a). The design optimization process was performed via 2D and 3D COMSOL® Finite Element Modeling (FEM) simulations, and is further discussed in the Methods section.

For each device, the measured scattering parameters were later converted into admittance parameters ($Y$). Due to pads and routing configuration, $Y_{12}$ is not impacted by parasitics to ground, and was used to extract the resonators' motional parameters. According to the data, the devices showcase unprecedented loaded quality factor ($Q_{3dB}$) and $k_t^2$ values in this frequency range (Table 2), while having their static capacitance ($C_0$) closely matched to 50 Ω around resonance. Fig. 3a reports the resonators along with their modeling, while the fitting parameters ($Q_{fit}$, $k_t^2$, $C_0$, and $f_s$) are listed in Table 2. The fitting was carried out with a simple Butterworth-Van Dyke (BVD) equivalent circuit model[7]. $Q_{fit}$ values are ohmic-loaded, and therefore embed any parasitic resistance due to routing, IDTs, and the contact resistance. The discrepancy between $Q_{3dB}$ and $Q_{fit}$ was attributed to an overestimation of the $k_t^2$ due to the presence of in-band spurious modes, which artificially lower $Q_{fit}$ values to match the maximum measured

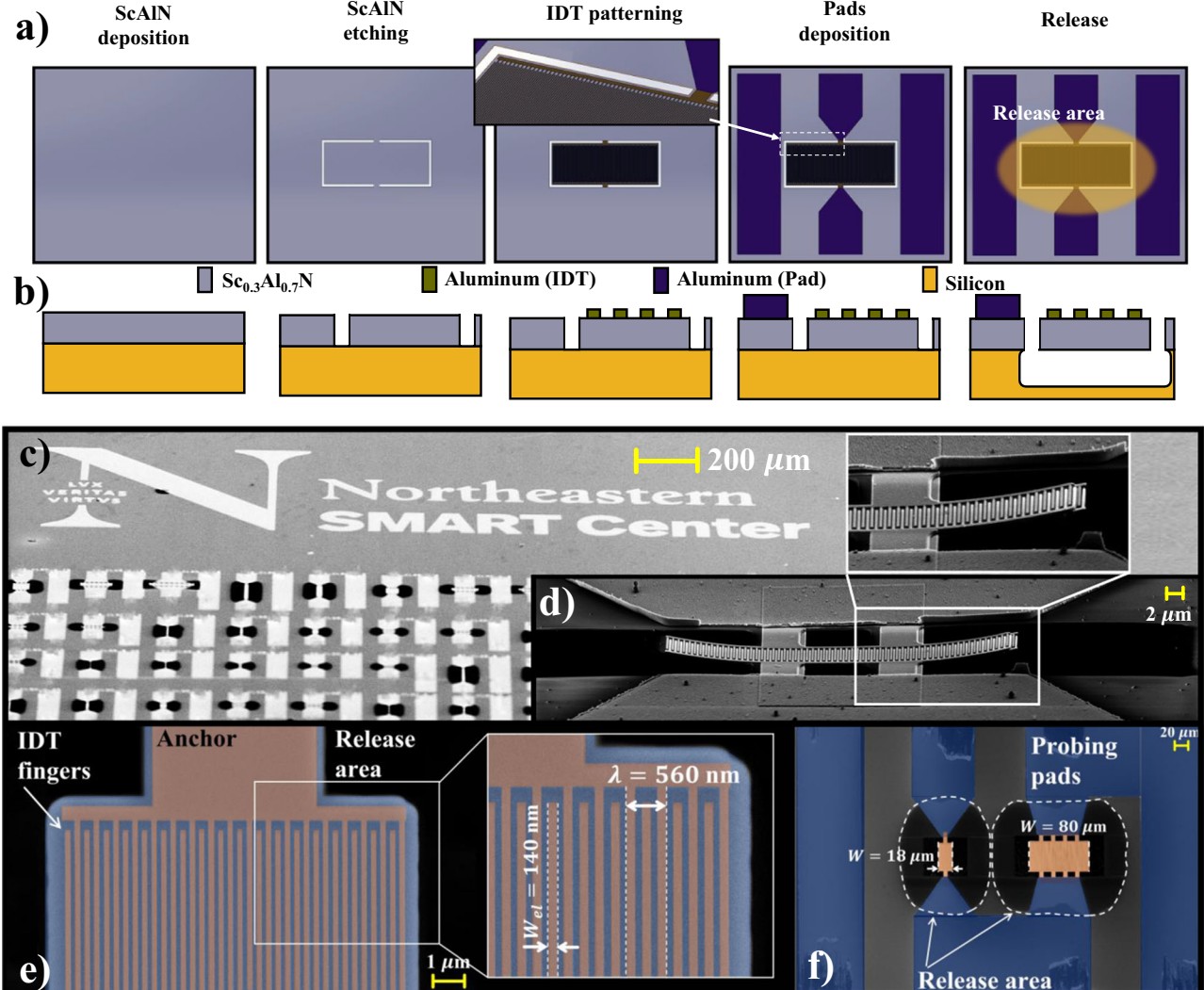

**Fig. 2 | Device fabrication and SEM micrographs. a, b** Micro-machining process flow adopted to fabricate the devices reported in this work in both a **a** 3D and a **b** cross-sectional views. Despite representing a single nanoacoustic resonator, this process was also utilized to fabricate the filters. **c**–**f** SEM pictures of fabricated resonators and filters, with their relevant dimensions highlighted. **c** provides an overview of different resonators on the same chip. In **d**, a device with a $\lambda$ of 933 nm and operating at 8.35 GHz is shown. The bending is due to residual stress-gradient in the film, quantified to be 0.9 GPa/$\mu m$. In **e**, a device with a $\lambda$ of 560 nm and operating at 10.2 GHz is shown, while **f** depicts a first-order ladder filter with center frequency of 9.17 GHz.

admittance. The outstanding metrics of the present devices directly translate into significantly higher $f_s \cdot Q$, $Q \cdot k_t^2$, and $f_s \cdot Q \cdot k_t^2$ Figures of Merit (FoM) products compared to literature results. The scatter plots of Fig. 3b–e provide a comparison between the resonators of this work with the most relevant ones proposed in the literature above 6 GHz. The plots include both non-lithographically and lithographically frequency-defined resonators, and a broad selection of material platforms regardless of CMOS compatibility[24,25,30,32,33,35,37,38,43,51–55].

The adoption of different metrics to evaluate the quality factor of a resonator (e.g. $Q_{3dB}$, $Q_p$ or anti-resonance $Q$, $Q_m$ or motional $Q$[7], and $Q_{max}$ or Bode $Q$[56]), some of which require significant data manipulation or de-embedding, along with the lack of standardized criteria for its extraction, significantly complicates any state-of-the-art comparison. To address this challenge, the authors decided to report the largest $Q$ metric presented in each of the the referenced publications. A clarification on the specific metric adopted by other authors in their works is reported in the caption of Fig. 3.

According to the data reported in Fig. 3, the devices presented in this work possess the highest $Q_{3dB}$ in the 6–20 GHz range among the cited publications. To offer a fair comparison between the devices described in this work and their state-of-the-art counterparts, both $Q_{3dB}$ and $Q_{max}$ are reported (Table 2). In Fig. 3b–e, the optimal performance region for the synthesis of RF filters for the mid-band 6G spectrum is highlighted in green.

As observable in Fig. 3a, the devices present in-band spurious modes between the resonance and the anti-resonance. Their presence is detrimental to the performance of filters, causing unwanted ripples in the pass-band and group delay distortion. These modes were identified as transverse excitations, which are well-known to originate in CLMRs[57], via FEM analysis. IDTs finger apodization techniques, first introduced in SAW resonators[8] to control and shape the distribution of acoustic energy in the resonator's active area, were experimentally demonstrated to successfully suppress spurious resonances in CLMRs[57] and other kinds of Lamb wave resonators[58,59].

## Nanoacoustic pass-band filters

To demonstrate the potential of the HD ScAlN CLMR platform, a first-order ladder filter topology was chosen for the synthesis of high performance mid-band filters. Figure 4a reports the $S_{21}$ scattering parameter response of a bank of nanoacoustic CLMRs-based filters operating in the 7–11 GHz range and fabricated on the same substrate. The filter response benefits from the same lithographic definition as

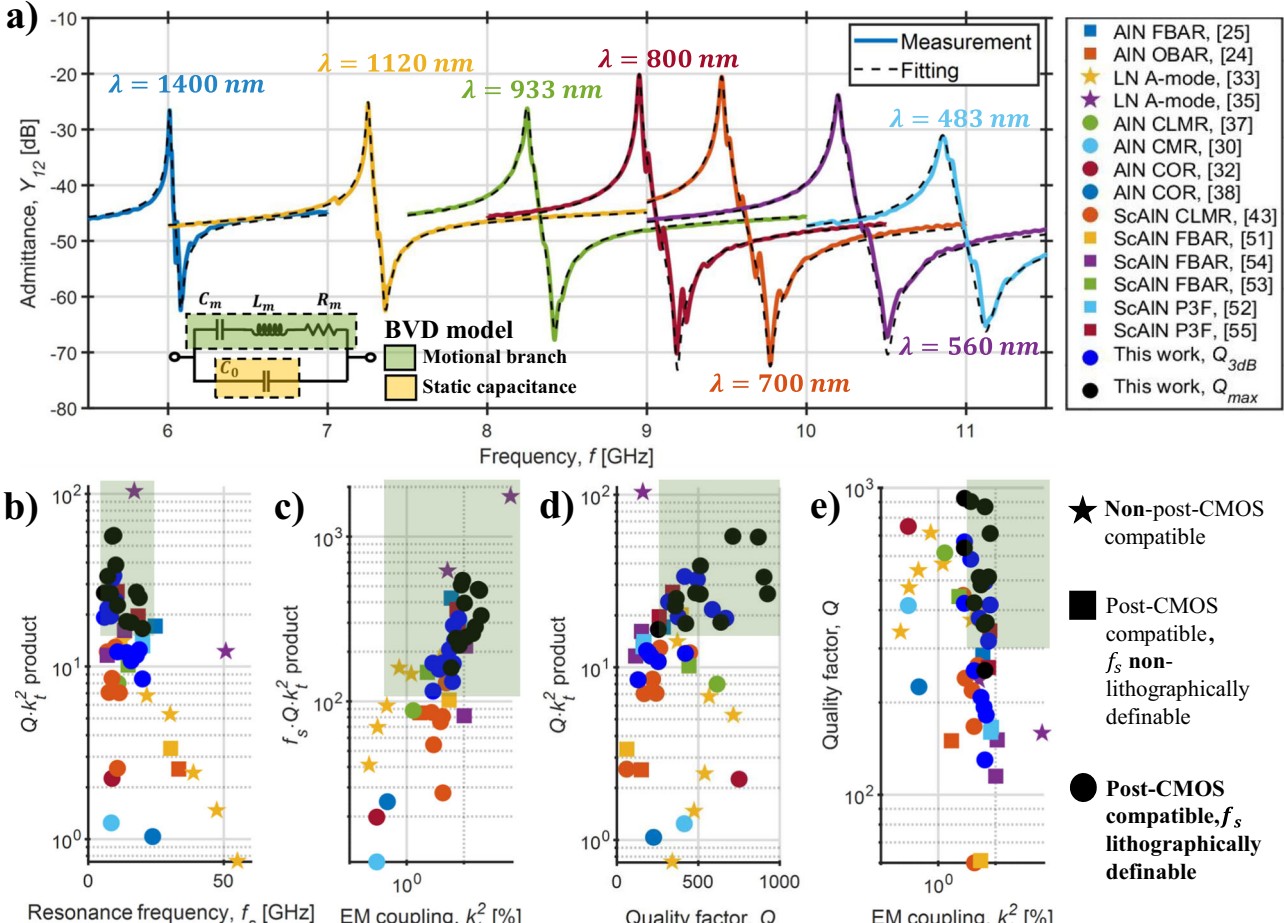

**Fig. 3 | Resonator admittance and scatter plots. a** Measured admittance $Y_{12}$ vs. frequency curves for showcase resonators from this work, along with their BVD fitting. The fitting parameters are shown in Table 2. **b–e** Scatter plots comparing the devices from this work with resonators in the literature above 6 GHz. The optimal performance region for the synthesis of filters for the 6G mid-band is highlighted in green. The considered frequency range is 7–20 GHz, while a $Q \cdot k_t^2$ product of 15 is

the considered minimum requirement to achieve acceptable values of filter insertion loss. A minimum $k_t^2$ of 4% is also assumed to ensure a required bandwidth of 400 MHz[4] at 20 GHz. Due to the lack of a generalized guideline for quality factor extraction, the maximum $Q$ retrieved from the referenced publications is reported. More in detail, motional $Q$ is used for[25,30,32,33,37,38,51,54], $Q_{3dB}$ for[35,43], $Q_{max}$ for[52,53], and $Q_p$ for[55].

the resonance frequency of the individual resonators. In this first demonstration, the filters were not designed to be 50 Ω-matched. To remove the return loss due to wave reflections in case of impedance mismatch, the filter responses were post-processed via Advanced Design System® (ADS). The aim of the de-embedding was to change the impedance terminations from the 50 Ω of the VNA to the values shown in Table 3, providing IL figures that come solely from the physical limitations of the nanoacoustic devices, i.e. the resonators' motional and parasitic resistances. The filter responses of Fig. 4 are

matched, while Fig. 4b shows the comparison of the unmatched (raw) data and the matched one for a showcase device. No inductors were used on ADS®, and therefore the matching process corresponds to a simple resistive change of impedance at the RF ports.

Finally, the metrics of the filters in Fig. 4a, i.e., center frequency ($f_c$), IL, Out-of-Band Rejection (OBR), $BW_{3dB}$, and fractional BW (FBW), are reported in Table 3, which also includes previously demonstrated devices from[43]. This aims to quantitatively demonstrate how improvements in film crystallinity and the micro-machining process substantially impact the primary metrics of these devices, thereby underlining the novelty of these results.

### Filter simulations and performance enhancement

The filter routing layout is reported in Fig. 4c. This probing configuration allows to independently extract the filter's and the resonators' S-parameters by means of GS and SG probes. This specific pad configuration was developed to extract the resonators' admittances for a-posteriori reconstruction of the filter $S_{12}$ response (cfr. Methods section). A first, third, and fifth order ladder filter were simulated with ADS® by employing the BVD-fitted representation of the extracted series and parallel resonators. Fig. 4d reports the filters' topologies together with their matched responses. As expected, an increase in the filters' order translates into larger IL and OBR specs. The $S_{12}$ responses of the filters, unlike the ones in Fig. 4a, are spurious-free. The reason

### Table 2 | Metrics of the resonators shown in Fig. 3

| $f_s$ [GHz] | $Q_{3dB}$ | $k_t^2$ [%] | $C_0$ [fF] | $Q_{fit}$ | $Q_{max}$ |
|---|---|---|---|---|---|
| 6 | 668 | 2.88 | 131.3 | 450 | 926 |
| 7.25 | 586 | 3.69 | 105.9 | 380 | 903 |
| 8.25 | 377 | 5.19 | 91.47 | 250 | 512 |
| 8.95 | 494 | 6.54 | 77.32 | 420 | 869 |
| 9.47 | 416 | 8.07 | 73.76 | 330 | 712 |
| 10.2 | 317 | 7.55 | 64.67 | 250 | 513 |
| 10.85 | 193 | 6.3 | 51.97 | 160 | 359 |

$Q$ and $k_t^2$ were computed as $Q_{3dB} = f_s / BW_{3dB}$ and $k_t^2 = \pi^2/8 (f_p^2 - f_s^2)/f_s^2$, where $f_s$ and $f_p$ are the resonance and anti-resonance frequencies, respectively. $BW_{3dB}$ represents the admittance's 3dB bandwidth. $Q_{max}$ is extracted as in[56].

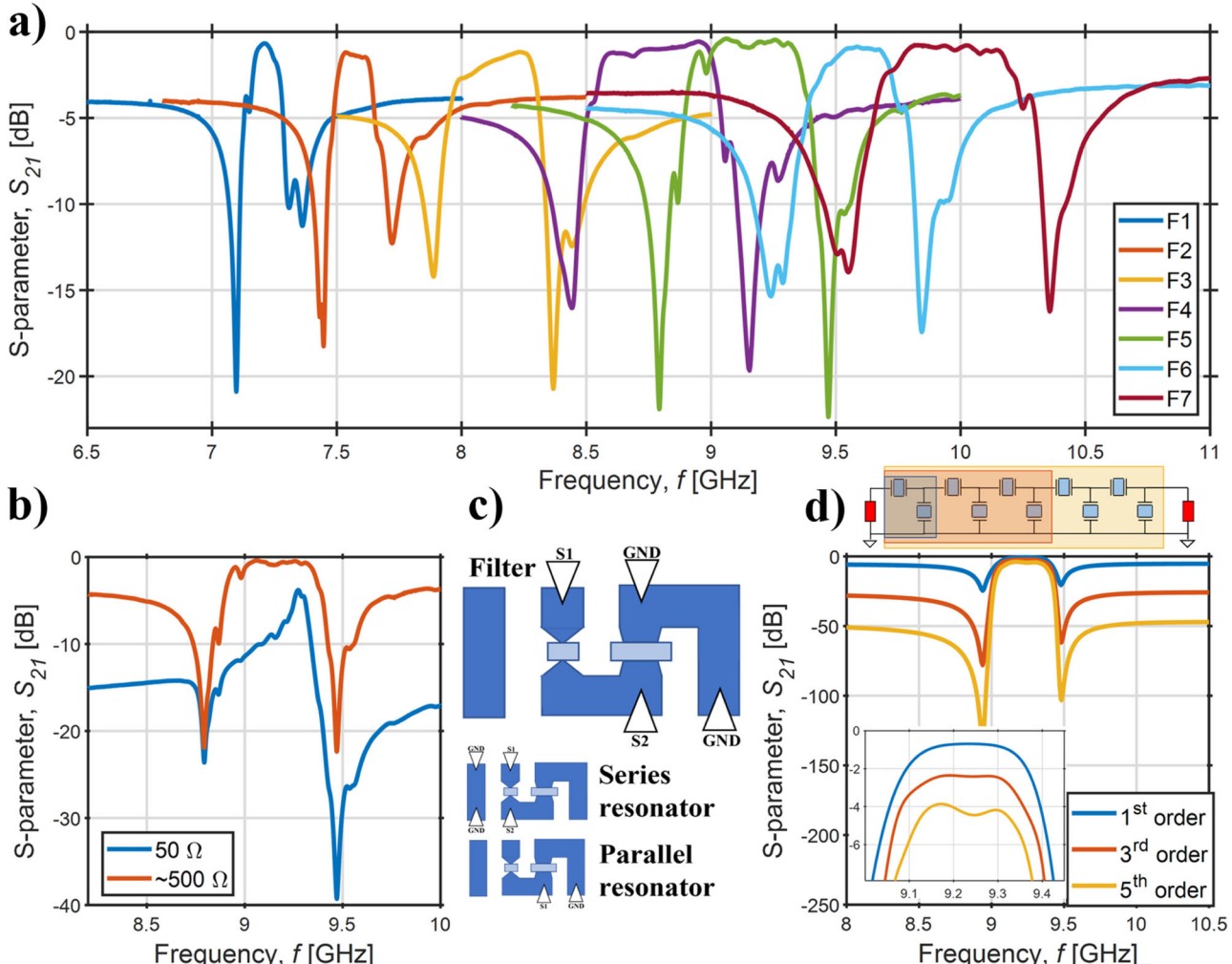

**Fig. 4 | Microacoustic ladder filters. a** Measured filter scattering parameter $S_{21}$ vs. frequency for a bank of nanoacoustic devices fabricated on the same chip and matched via ADS® (see Table 3). In **b**, a comparison between unmatched (raw) and matched responses. In **c**, the pad structure is reported, together with the probing configurations to independently measure the filter and the series and parallel resonators. In **d**, ADS simulated and matched 1st, 3rd, and 5th order F5 filters are shown, starting from the measured and single-mode-fitted resonator responses (see Methods). The notch depth of the 3rd and 5th order filters is exacerbated by the employment of the simple BVD model to describe the resonator response. In a real case, the dielectric losses inherent to the piezoelectric layer[7] would cap the notch depth. Such losses were not taken into account here to be consistent with the BVD fitting used throughout this work.

lies in the utilization of the single-mode fittings of the building blocks resonators instead of the measured data. In fact, the impact of spurious modes becomes more evident when the filter order is increased, and design techniques such as IDT fingers apodization must be used to suppress them before synthesizing higher order devices.

**Compact dimensions for handset applications**

The active area of the fabricated filters is solely constituted by the active areas of the series and parallel resonators. Given their extremely small pitch, the overall filter dimensions are very limited, ranging between 0.0015 mm² for F1 and 0.00048 mm² for F7. As the devices are matched to impedance values around 500 Ω, a 10-fold active area increase has to be considered when synthesizing filters matched to the 50 Ω standard. Nevertheless, the form factor remains contained and competitive with respect to state-of-the-art solutions. In addition, greater area reduction can be achieved when scaling such filters to the higher end of the 7–20 GHz range due to a reduction in the horizontal acoustic wavelength.

When considering interconnects, which are as first approximation equal for all the devices, with 0.432 mm of width and 0.22 mm of length, the total area increases to 0.095 $mm^2$. This footprint demonstrates the suitability of this technology for its integration into handsets. Naturally, the occupiedarea can be further reduced by removing the additional pads introduced to individually probe series and parallel elements within the filter structure.

In conclusion, the current work proposes a technology platform able to solve the need for compact filtering devices in the 7–20 GHz range. This technology, namely highly-doped ScAlN Cross-Sectional Lamé Mode Resonators, enables the synthesis of lithographically definable, post-CMOS compatible, high-performance, and compact RF filters in the 7–20 GHz range which would fit into cellular radios for the next generation of mobile networks.

Thanks to a fabrication process compatible with foundry-standard 200 mm wafers, multi-frequency lithographic definition on the same chip with a single mask, and the general ease of fabrication compared to other commercial technologies, this work presents an extremely promising approach for cost-effective mass-production of acoustic filters for 5G and beyond applications.

## Methods

### ScAlN thin-film deposition optimization

Highly c-axis oriented ScAlN films are nowadays a fundamental requirement for the fabrication of high-performance, post-CMOS compatible resonators. More specifically, AOGs have been identified as

**Table 3 | Main metrics for the filters of Fig. 4a**

| Filter | $f_c$ (GHz) | min $IL$ (dB) | $BW$ (MHz) | $FBW$ (%) | $OBR_M$ | $\overline{Z}_M$ (Ω) |
|---|---|---|---|---|---|---|
| F1 | 7.2 | -0.67 | 121 | 1.7 | -4.6 | 500 |
| F2 | 7.57 | -1.19 | 156 | 2.1 | -4.55 | 476 |
| F3 | 8.15 | -1.17 | 350 | 4.3 | -5.5 | 441 |
| F4 | 8.84 | -0.56 | 489 | 5.53 | -5 | 453 |
| F5 | 9.17 | -0.39 | 469 | 5.11 | -4.95 | 513 |
| F6 | 9.6 | -0.86 | 345 | 3.6 | -4.9 | 511 |
| F7 | 9.96 | -0.76 | 567 | 5.7 | -3.8 | 508 |
| From[43] | 7.06 | -3.6 | 90 | 1.3 | -10.3 | 120 + j168 |
| From[43] | 10.4 | -3.9 | 244 | 2.35 | -11.6 | 141 + j287 |
| From[43] | 11.15 | -3.62 | 423 | 3.78 | -11.4 | 200 + j255 |

The metrics refer to the matched filters. The reported *IL* represents the minimum in-band insertion loss. The *OBR* is computed by averaging the values of $S_{12}$ measured 1 bandwidth away from the center frequency, and the reported matching impedance is calculated by averaging the impedances of the two ports. In the same table, the filters from[43] are also reported.

a source of performance reduction in acoustic devices, both in terms of $Q$ and $k_t^{2[50]}$. To avoid Figure of Merit degradation, deposition conditions must be finely tuned to mitigate AOGs formation, achieve excellent crystallinity, and maintain residual stress levels within the failure tolerance of the suspended plates[46,60,61].

ScAlN films were deposited by reactive sputtering, either using a single AlSc alloy target, or by co-sputtering from separate Al and Sc targets. In this work, a 12" $Al_{0.7}Sc_{0.3}$ casted target to supply the Al and Sc atoms was used a source of the metallic material. Sole nitrogen was adopted as carrier gas (i.e., no Argon), as it was found to improve the film's crystallinity and reduce AOGs formation[48,60,62–64]. The optimal gas flow and target applied power were set to 30 sccm and 5 kW, respectively. The chuck temperature was set to 300 °C, a value that is compatible with standard CMOS back-end processing.

## FEM simulations
The devices' design validation was performed via COMSOL® 2D and 3D Finite Element Analysis (FEA). The ab-initio equations from[40] for the stiffness and piezoelectric coefficients and for the dielectric constant were employed, as in[44], to define ScAlN material properties as a function of doping.

As for the top electrode, aluminum (Al) was chosen for its low resistivity, although it provides the lowest $k_t^2$ among all the possible top electrodes given the small difference in its acoustic impedance compared to ScAlN's[44]. Moreover, its very low density allows to achieve higher frequencies for a given pitch compared to denser metals such as Mo, Pt, and W, thus relaxing the lithographic contraints. An IDT thickness of 95 nm was chosen via FEM simulations as the one maximizing the electromechanical coupling for the resonator with the optimal $h/\lambda$ ratio of 0.4, as shown in Ref. 44.

COMSOL® FEA was used to evaluate the expected $k_t^2$ and $f_s$ of the resonators for the purpose of fabricating filters. Moreover, the resonators of Fig. 3a have been a-posteriori simulated in order to take advantage of the real fabricated geometrical dimension. In Fig. 5a–b, a comparison between measured results and FEM simulations is provided in terms of electromechanical coupling (a) and resonance frequency (b) as function of the $h/\lambda$ ratio. In the plot, each data point is the average between six or more devices with the same $\lambda$ fabricated in two different batches. An error bar showing the maximum and the minimum value for each geometrical ratio is also provided. As can be observed, there is good agreement between FEM simulations and experimental data. In terms of $k_t^2$, the measured devices showcase a certain variability, which can be explained by fabrication non-idealities. Therefore, more process optimization must be performed to decrease the device variability. However, the observed variability of the resonance frequency is low.

## Device fabrication
Standalone resonators and nanoacoustic filters in the 6 to 11 GHz range were fabricated on the same 200 mm Si substrate with a simple micromachining process (Fig. 2). First, the 280 nm $Sc_{0.3}Al_{0.7}N$ film was deposited. The plate geometry was then patterned by dry-etching the piezoelectric film. For that purpose, an Oxford Inductively Coupled Plasma (ICP) Reactive Ion Etching (RIE) tool was used. To etch the hard ScAlN, a recipe with high Argon and high power was selected, together with Chlorine chemistry ($Cl_2$ and $BCl_3$ gases). To improve the plate edges, an oxide mask deposited via Plasma Enhanced Chemical Vapour Deposition (PECVD) was utilized. It was etched via RIE in fluorine chemistry and stripped in 49% hydrofluoric acid. After this step, electron-beam lithography was performed to define the thin finger electrodes ($W_{el}$ = 120–350 nm), followed by 95 nm aluminum thermal evaporation and lift-off. The pads were fabricated in a second step via direct writer lithography, aluminum sputtering and lift-off. In a foundry, where cost-effectiveness and throughput are the main concerns, the two steps could be combined with an industrial-grade extreme-UV stepper. As a last step, the devices were released via $XeF_2$ isotropic etching.

## Device temperature coefficient of frequency
The Temperature Coefficient of Frequency (TCF) of the present devices was investigated by changing the probe station's chuck temperature from 25 to 145 °C in steps of 20 °C. Simultaneously, the frequency shift of multiple devices was recorded. The TCF was extracted via linear fitting of the measured shifts and averaged between the different devices, resulting in -36.1 ppm/K. The obtained result shows an increase of the TCF compared to similar AlN devices[65] and that further optimization of such value needs to be performed. Possible solutions include the use of a silicon dioxide compensation layer[65].

## Bandwith enhancement
The filter $BW$, directly proportional to the resonators' electromechanical coupling, can be further increased in LFE CLMRs by increasing the Sc-doping level, or employing higher acoustic impedance metal electrodes, such as Pt, W, or Mo. In the first case, the $k_t^2$ increases because of an increase in the piezo coefficients and of a decrease of equivalent stiffness, according to the following equation[7]:

$$k_t^2 \propto \frac{e^2}{\epsilon \cdot C} \qquad (1)$$

where $e$ is the piezoelectric coefficient, $\epsilon$ is the dielectric constant, and $C$ is the equivalent stiffness. The electromechanical coupling enhancement deriving from different IDTs metals derives from the increased mismatch in acoustic impedance between the piezo and the electrode, resulting in a better confinement of the resonant mode inside the resonator cavity[44]. Figure 5c shows the COMSOL® simulated $k_t^2$ vs. Sc concentration in the piezoelectric film for an LFE CLMR with $h/\lambda$ = 0.4.

Moreover, fabrication complexity can be traded off in terms of $k_t^2$ increase by adding a continuous bottom electrode, therefore realizing the Floating Bottom Electrode (FBE) CLMR, demonstrated to reach higher couplings in a wider $h/\lambda$ range of values[44].

## Filter reconstruction and higher order simulation
By exploiting the filters' pad structure shown in Fig. 4c, the resonators' frequency response was extracted. Figure 6a–b shows the admittance curves of the resonators raw data and the ones of the devices when the extra parasitics introduced by the filters pad structure are removed for F5, taken as a showcase. Figure 6c depicts the $S_{12}$ de-embedded scattering parameters of the series and parallel resonators and the raw data of the corresponding filter. Figure 6 shows the excellent agreement of the reconstructed, unmatched filter with the measured results, validating the model. Finally, by employing the single-mode-fitted

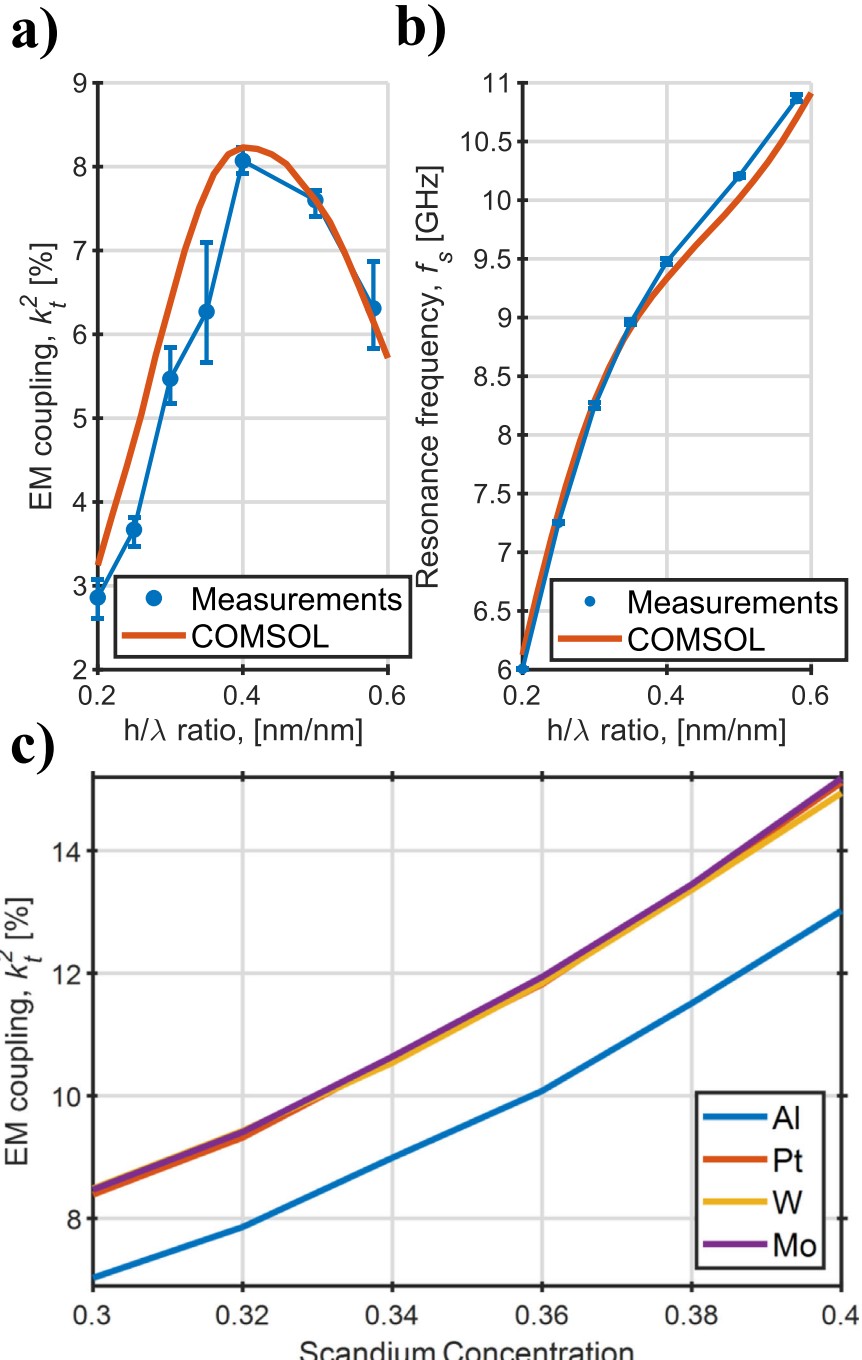

**Fig. 5 | FEM simulations and agreement with experiments. a, b** A-posteriori 2D COMSOL® simulations of the devices shown in Fig. 3 and agreement with the experimental data in predicting $k_t^2$ **a** and $f_s$ **b.** The error bars show the maximum and minimum measured value per data point. In **c**, COMSOL® simulations of $k_t^2$ as function of Sc-doping of the piezoelectric film for different metal electrodes are provided. The simulated device has an $h/\lambda$ ratio of 0.4. The IDT thicknesses are 95, 30, 40, and 70 nm for Al, Pt, W, and Mo, respectively. This optimal values were found via COMSOL® FEM.

frequency responses for the de-embedded resonators of Fig. 6a–b and by slightly adjusting the parallel resonator's resonance frequency to exactly match the series', a 1st, 3rd, and 5th order filters were simulated with ADS® (Fig. 4d). After that, the filters were matched to impedances around 500 Ω. This shows the expected improvement in *OBR* when increasing the number of ladder stages, and the next expected steps in the technology development.

## Filters power handling
The present paper aims to showcase on-chip multi-frequency high-performance passband filters based on ScAlN CLMRs. Nevertheless,

given the early stage of the technology development, the proposed filters still require a considerable amount of development to meet commercial specifications. Among those, power handling is particularly critical for any RFFE. In Fig. 6e, the ADS® $S_{21}$ matched scattering parameters of a measured filter are reported as a function of the applied input power $P_{in}$. The filters experience 1 dB transmission characteristics degradation in the passband after the application of 9 dBm (Fig. 6f). The devices' irreversible breakdown mechanism is identified as electromigration in the aluminum IDTs due to the large alternated currents passing through the devices at their resonance (Fig. 6g), especially due to the low motional resistance. Possible

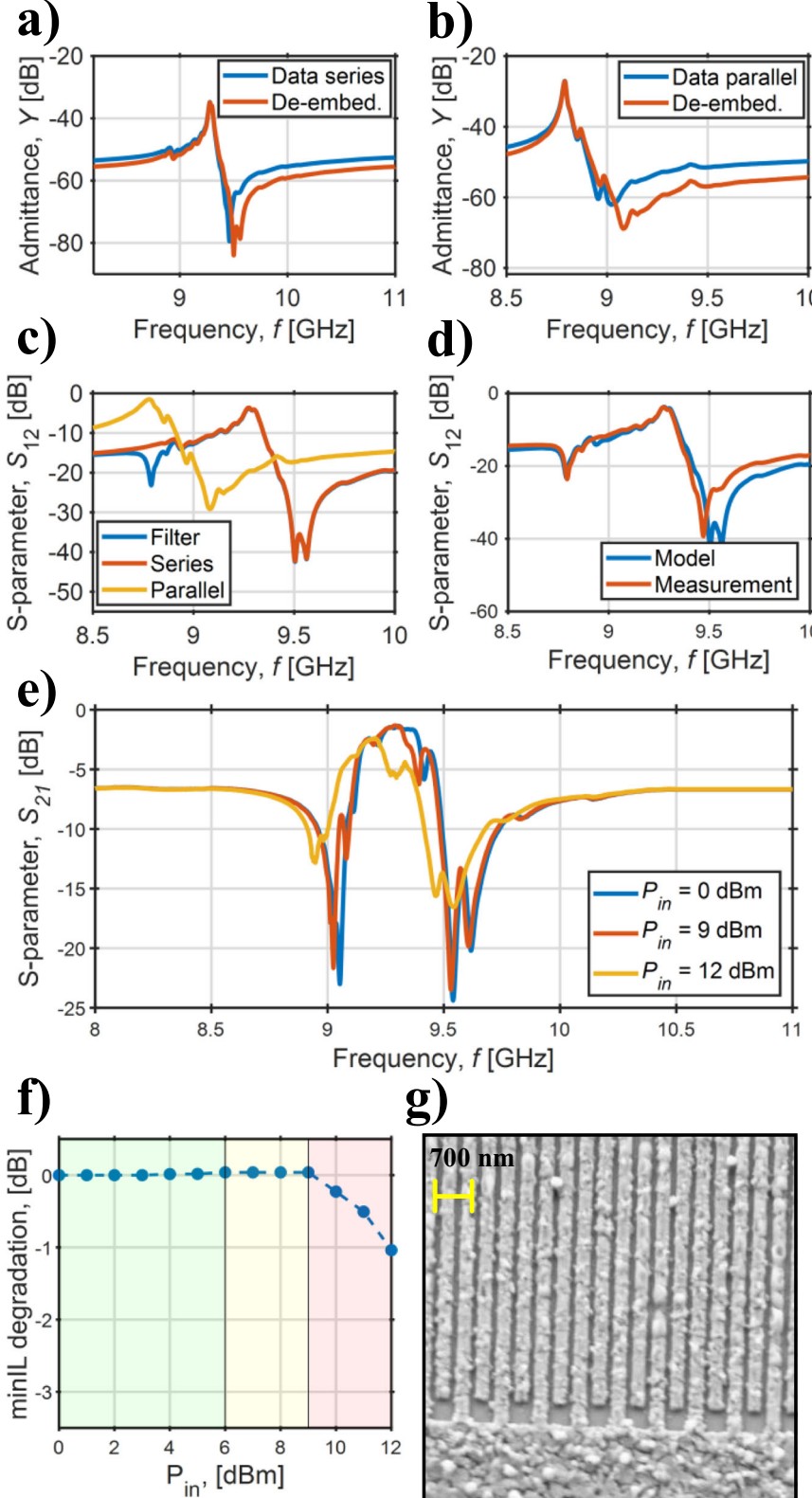

**Fig. 6 | Filter analysis. a, b** Measured and de-embedded admittance responses of series and parallel resonators of F5 (Fig. 4a), **c** $S_{21}$ of the de-embedded resonators and of the filter F5, and **d** comparison between measured and reconstructed filters from the responses shown in **c**. The discrepancy in the responses likely derives from the impact of pads and interconnects' parasitics. **e**) Measured $S_{21}$ scattering parameters of a showcase filter as function of the input power. In **f**, the minimum *IL*

degradation is plotted as function of the input power. The green region is the linear one, while the yellow indicates a input power region which causes a reversible *IL* degradation. The power levels in the red region cause irreversible device damage. In **g**, an SEM micrograph of the IDT fingers of the series resonator of the filter after experiencing 12 dBm of applied power. Electromigration is identified as IDTs breakdown mechanism.

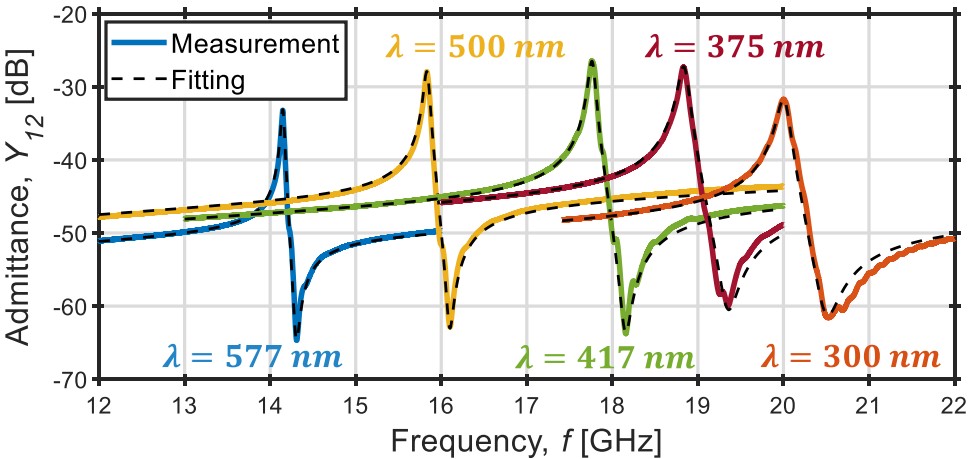

**Fig. 7 | On a further CLMR scaling.** Measured and BVD fitted admittance vs. frequency responses of resonators operating in the 14–20 GHz range, demonstrating the scaling capabilities of the LFE CLMR technology.

solutions include the employment of AlSiCu or AlSicu-Ti-AlSiCu metal stacks instead of pure Al in order to mitigate the impact of electromigration, as commonly done in CMOS foundries. Moreover, the power handling is expected to notably improve with the use of 50-Ω-matched filters, since the lower impedance matching requires more IDT fingers, allowing to have less current per single electrode, thus increasing the power level that generates the critical current density responsible for the devices' breakdown.

### Scaling to the higher end of the spectrum

The Results section highlighted resonators and filters up to 11 GHz of operation. Nevertheless, since the ultimate goal is to extend the frequency of operation up to and beyond 20 GHz, further reduction of the film thickness and in-plane dimensions is required. Such scaling poses several challenges, specifically on the piezoelectric layer growth side and on the increased lithographic resolution. In order to demonstrate the capabilities of the current technology platform, a 150 nm film was deposited by employing the same sputtering recipe, yielding a 2.7° FWHM. Moreover, LFE CLMRs in the 14–20 GHz range were fabricated on it (Fig. 7) employing the same fabrication process previously described, with the only difference being an IDT thickness of 43 nm, as that value maximizes the coupling for the CLMR with the optimal $h/\lambda$ ratio with the used material stack. The devices showcase $Q$ and $k_t^2$ values with only minor degradation compared with the 6–11 GHz ones. In fact, ectromechanical couplings in excess of 6% in the 18–20 GHz range were measured, as well as $Q_{3dB}$ as high as 421 at 14 GHz, and above 200 up to 18 GHz[66]. The BVD fittings are provided, and the devices' metrics were added to the scatter plots of Fig. 3b–e. The performance degradation is explained by the lower quality of the piezoelectric layer, already seen by the authors to greatly impact both the $Q$ and the $k_t^2$ of resonators[45]. In[43], in fact, the same authors show $Q_{3dB} \cdot k_t^2$ figures of merit in the 7–11 GHz range that are comparable to the ones of the 14–20 GHz presented in this section, while employing a piezoelectric layer with similar crystallinity. In Ref. 45, by simply adopting a substrate with improved FWHM and surface roughness, the measured figures of merit are increased by a 3-fold factor. Therefore, further thin film sputtering development is necessary to bring thinner film quality on par with the 280 nm one. Still, the present results highlight the capabilities of the present technology to cover the higher end of the 7–20 GHz spectrum, and that the reduced in-plane dimensions and layer thicknesses do not compromise the resonator performance. In terms of mass-production of these devices, the minimum feature size adopted in this work (75 nm) is manufacturable by state-of-the-art extreme ultraviolet stepper tools.

## Data availability

The S-parameter files of the resonators and filters of this work have been deposited in the Zenodo database with https://doi.org/10.5281/zenodo.8048267. Any other data is available upon request.

## Code availability

The MATLAB® codes used to generate the figures are available upon request.

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

## Acknowledgements
This work was supported by the Defense Advanced Research Projects Agency (DARPA) under Contract HR0011-22-C-0088, M.R. We would like to thank the staff of the Kostas Nanotechnology Laboratory at Northeastern University and the Center for Nanoscale Systems (CNS) at Harvard University.

## Author contributions
G.G., L.C., and M.R. conceived the concept and initiated the research. G.G. and P.S. performed the ScAlN thin-film optimization. G.G. fabricated the devices. G.G. and L.C. analyzed and processed the data. L.C. and M.R. supervised the research. G.G. wrote the paper body while all authors contributed to the manuscript's polishing.

## Competing interests
The authors declare no competing interests.
