## [Peer Review File · Nature Communications]

Compact and wideband nanoacoustic pass-band filters for future 5G and 6G cellular radiosREVIEWER COMMENTS

Reviewer #1 (Remarks to the Author):

This manuscript presents important advances in front-end microelectronic components enabling advances in front-end hardware for next generation wireless communication. Scandium-doped AlN is the front-runner for adaptation in commercial, industrial, and defense applications, and as such, this work is of great interest to the broader RF community. The authors have performed important material optimization in combination with an advantageous resonance mode and excitation necessary for maintaining high resonator/filter performance with frequency scaling to 20 GHz relevant to 5G/6G implementation.

The work and results reported are sound. To bring the quality of the paper to the level of Nature Communications, some additions and clarifications need to be addressed:

1. Given the ADS de-embedding, matching, and fitting performed in the analysis of the resonators/filters in this work, the authors should post their raw data in an open repository and make it available for the public to perform their own analysis/fits if desired. The link to this repository can be provided in the paper.
2. Post-CMOS compatibility of ScAlN is mentioned several times. Are there references that can fortify this claim? The mention of 300C deposition is helpful, and I understand that there is interest and investigation among foundries to allow Sc, but am not aware of any CMOS foundry currently implementing ScAlN, nor reports that it is safe for CMOS functionality.
3. Pg. 3, Col 1: "...precision required for such steps for below-6 GHz spectra..."
A reference to support the Angstrom-precision required would be very helpful. This also necessitates a discussion on how CLMR devices with LFE are more robust to the fabrication variations than FBARs. What is the precision required for 5G/6G? What does that translate to in IDT pitch, electrode width, electrode thickness, piezo film thickness, etc.? This analysis should be added to the manuscript from both fundamental mode shape (simulation) model as well as from intra-chip measurements on identical devices and chip-to-chip variations.
4. The authors discuss the non-linear dependence of k_2 on doping concentration, stating on Pg. 4, Col 1 "Because of the non-linear change.. the maximum electromechanical coupling enhancement is achieved for high doping levels." Later in the Results section, reference to cubic/quadratic dependence determined through DFT is mentioned and referenced. However, the strict correlation to k_2 with nonlinear dependence on doping concentration isn't clear. Can the authors elaborate on the physical mechanisms (stress, etc.) behind this? It would also be good to refer to Methods Fig. 9 in that discussion.
5. In Results: Highly-doped ScAlN section, it would be good to add comments on 8" and larger substrate deposition capabilities, uniformity, CMOS-compatibility, etc. Also in that same section, Fig. 2 needs some improvement. Specifically, in Fig. 2c it is very difficult to get anything useful out of the image. The AOG does not resolve well, particularly in print. It would be better to change this to an AFM scan with color map to highlight the defect density the authors are trying to demonstrate. AFM measurements should also replace Fig. 7 in Methods. Fig. 2d similarly does not show much. A TEM scan rather than SEM scan would be much more informative and provide better contrast. Ideally, TEM with EELS to show composition should be performed.
6. In Fig. 2b and Fig. 7, how does the FWHM of the authors' films compare with that of standard ion-assisted sputtered AlN, epitaxial AlN, epitaxial ScAlN, etc.? A frame of reference for the measured FWHM would be very useful.
7. One known challenge for ScAlN, particularly in the case of heavily doped films such as that in this work, is scaling film thickness down while maintaining high k_2 . The k_2 dependence on h/λ (e.g. Fig. 8a) tells us that scaling to 20 GHz will still require further thinning of the film. Discussion on this and on any limits/challenges/state of the art in ultra-thin-film ScAlN should be included in the paper.
8. In Fig. 4, it is unclear which of these devices are standalone resonators and which are filters. It becomes more evident after Fig. 6c, but for clarity, please include labels to this effect in the figure directly. The authors should also specify in the caption the frequencies of operation for the devices shown, and comments on the bowing shown in the long (lower frequency?) resonator.

9. In the section on Standalone Nanoacoustic Resonators, Pg. 6, it is worth discussing the choice of metal and thickness, especially considering the nm dimensions with frequency scaling. Ultimately it is the transducer resistance and mass loading that will limit scaling. What does that look like up to 20 GHz in the LFE CLMRs?
10. In the same section, Pg. 6 and Fig. 5(b-e), the scatter plots of various devices indicate "similar devices all fabricated on the same chip in the same process flow." Does this indicate spread in performance for identical designs, or rather a range of different designs with intentionally varying performance? As noted in reviewer comment #3, it is worth discussing fabrication variations and statistics for identical devices across chip and chip-to-chip.
11. In Fig. 5, the green highlighted regions indicate optimal performance for 6G filter synthesis. How are these performance metrics calculated or selected?
12. Fig. 6(d) should include an inset zoomed in on the passband. Also, what is the impedance matching for each order filter?
13. In the section on Compact Dimensions for Handset Applications, Pg. 7, How are the number of IDT fingers and overall resonator area chosen? What is the size limit of each device based on released films (we see the bowing/curling in Fig. 4)?
14. A discussion of ferroelectricity is not included in the paper. At 30% Sc doping, one would expect FE properties. Is that the case here? If so, the Methods section should also include PE loop measurements. This also prompts a discussion of power handling in the main manuscript, especially for thin films at higher frequencies.
15. In the Methods section on FEM Simulations, Pg. 7/8, can the authors explain the large discrepancy between DFT and measured relative permittivity from a physical/material perspective?
16. Comments/measurements on temperature dependence and stress gradients would be helpful in the Methods section.
17. Some typos/grammar:
 - a. Abstract: "...(5G FR-2) spectrum" should be changed to spectra
 - b. Abstract: "yet-to-explore" should be "yet-to-be-explored"
 - c. Intro: "mole of transmitted data" ?
 - d. Pg. 2: "devices' Radiofrequency"
 - e. Pg. 2: "it is foreseen to be refarmed in 6G..." ?
 - f. Pg. 2: "commercial products"
 - g. Pg. 2: first introduction of IDTs – reader may not know the jargon of "transducers/transduction"
 - h. Pg. 2: first introduction of "the parasitics" – needs to be clarified for general reader
 - i. Pg. 4: first instance of AOG in column 1 not explained until column 2.
 - j. Pg. 4: "devices built with this film as active layer" should be "as the active layer"
 - k. All instances of lithographic "tunability" should be changed to lithographic "definition."
 - l. Pg. 6: "...Q3dB has been chosen as metric..."
 - m. Pg. 6, section on Nanoacoustic pass-band filters: Several references to Table 2 should be changed to Table 3.
 - n. Pg. 7: "...In the same table, also the filters from [38]..." language grammar revision
 - o. English in last sentence of conclusion
 - p. Table 3 caption: sentence ending in "one"
 - q. Pg. 9 first paragraph: sentence ending in "one"
 - r. Methods section "Devices' Fabrication" should be changed to Device Fabrication
 - s. Methods section on Bandwidth enhancement, last line "in a wider h/lambda values"

Reviewer #2 (Remarks to the Author):

The authors report LFE CLMR's in the range of 6-11 GHz fabricated on the same chip in a 30% Sc doped ScAlN film. The results reported by the authors are indeed a giant leap forward for ScAlN Lamb wave resonators in obtaining high coupling, Q and FOM at these frequencies. In addition to some minor typos in the paper the authors need to better qualify their claims and what is still a considerable amount of research that will be required for the reported technology to be appropriate for cellular handset filters. If the authors can properly contextualize the limitations of the current technology in a revision, I would recommend publication in Nature Communications.

Major comments:

- 1) The authors claim the reported technology is appropriate for cellular phone duplexers that isolate the transmit and receive paths. In such a case the filters must be able to handle the full transmit power on order of 0.1 to 1 W. While the small, suspended resonators reported here are unlikely to approach these values, 1 dB compression point is very quick to measure and at a minimum should be reported to properly contextualize the further work that will be needed for the technology to be applicable to cellular handsets.
- 2) When the size of the filters is discussed on page 7, the reports are a bit misleading. In order to be matched to 50 ohms the area will need to be increased by 10 fold. This should be acknowledged.
- 3) The spurs in the resonators and filters would not be acceptable in a cellular phone filter. The group delay distortion would corrupt OFDM signals and would likely not be constant with input power. Can the authors acknowledge this and point to some approaches that might be used to obtain a response that is free of spurs like that seen in commercial BAW and SAW resonators/filters?
- 4) It is not clear from the paper how the simulated filter responses in Fig. 6 (d) were arrived at. The authors make it sound like they measured two resonators and then used the measurement results to construct higher order filters in ADS but I don't think that can be the case. The simulations seem to utilize spur free resonators but none of the measured resonators or 1-stage filters fit this description. Furthermore, the spurs will have a more degrading impact on filter performance for higher order filters with more stages. The authors should better describe the method used to obtain the resonator models used in the filter synthesis.
- 5) The authors tout the applicability of this technology from 7-20 GHz but show results from 7-11 GHz. It remains speculative that the frequency of this technology can be scaled to even higher frequencies while maintaining adequate performance and this should be acknowledged.

Minor comments:

- 1) In Fig. 7 I believe, based on the text narrative, that the left most image uses a 30 sccm N₂ gas flow. The figure 7 caption, however, shows the best film (left most) was achieved with a 60 sccm gas flow. The authors should verify and correct if needed.
- 2) The final paragraph in the methods makes reference to a Fig. ??a.
- 3) The notch depths in Fig. 6 d don't seem realistic. Such performance is not achieved for lower frequency 5-stage ladder filters using resonators with much larger FOM. Can the authors comment on this? Could something else limit the notch depth in a practical filter?
- 4) On page 3, the authors state "low modulus of the d₃₁ piezoelectric coefficient". I recommend changing "modulus" to "value".

Boston MA, USA
July 20th, 2023

We, the authors, would like to express our sincere gratitude to the reviewers for their constructive critiques. We acknowledge that the concerns they highlighted have played a significant role in refining and enhancing the quality of the present manuscript. Their dedication in reading, comprehending, and striving to uplift the quality of our work does not go unnoticed, and for this, we are profoundly thankful.

In response to their feedback, we provided detailed explanations to each comment, which were followed by the necessary revisions or additions to our manuscript. The changes implemented aim to thoroughly address and rectify all the issues raised, ensuring the integrity and quality of our work.

Sincerely,
The authors

Reviewer Comments

Reviewer #1 (Remarks to the Author):

This manuscript presents important advances in front-end microelectronic components enabling advances in front-end hardware for next generation wireless communication. Scandium-doped AlN is the front-runner for adaptation in commercial, industrial, and defense applications, and as such, this work is of great interest to the broader RF community. The authors have performed important material optimization in combination with an advantageous resonance mode and excitation necessary for maintaining high resonator/filter performance with frequency scaling to 20 GHz relevant to 5G/6G implementation. The work and results reported are sound. To bring the quality of the paper to the level of Nature Communications, some additions and clarifications need to be addressed:

The authors would like to thank the reviewer for the encouraging words and to their precious comments that helped us in better contextualizing our results, being clearer to a broader spectrum of readers, and to highlight the limitations of our technology and the research work that must be made in order to bring it to the commercial level.

1. Given the ADS de-embedding, matching, and fitting performed in the analysis of the resonators/filters in this work, the authors should post their raw data in an open repository and make it available for the public to perform their own analysis/fits if desired. The link to this repository can be provided in the paper.

Thanks for the comment. We uploaded the data on Zenodo for them to be available to the reader. The DOI is <https://doi.org/10.5281/zenodo.8048267>. We also added a data availability section reporting the link to the repository.

2. Post-CMOS compatibility of ScAlN is mentioned several times. Are there references that can fortify

this claim? The mention of 300C deposition is helpful, and I understand that there is interest and investigation among foundries to allow Sc, but am not aware of any CMOS foundry currently implementing ScAlN, nor reports that it is safe for CMOS functionality.

According to the author's knowledge, post-CMOS compatibility of sputtered nitrides has been long speculated due to the relatively low deposition temperatures of thin piezoelectric films (300-400 C) and is generally accepted within the community. According to our literature review, few works in the past years demonstrate integration of AlN on CMOS SOI platforms. Nevertheless, the authors believe that the lack many of experimental data and monolithic integration for RF front-ends is mainly due to technological constraints (e.g., lack of >8" tools for sputtering deposition, dedicated lines etc), large costs for experimentation (especially for non-corporate research), and yields (heterogeneous integration leads to lower overall yields). With respect to this last point, we believe that this paradigm will change due to more stringent requirements for 5G and 6G key parameter indicators. In other words, yields could be traded-off for parasitics reduction in order to enable higher data rate and bandwidth, especially at higher frequencies.

Given the similarities in tools and deposition conditions between AlN and ScAlN, for the purpose of this work we strongly believe that the current literature supports our claims, as ScAlN is always referred as being post-CMOS compatible, even in paper titles. To strengthen our thesis, we added the following sentence regarding AlN-CMOS monolithic integration on page 2:

"Monolithic integration of AlN acoustic resonators for filters, oscillators, fingerprint sensors, energy harvesting, and RF applications was successfully demonstrated [20, 21]. Despite the lack of available literature, post-CMOS ScAlN-compatibility is widely accepted within the community, due to the similarity in deposition conditions and machining between AlN and ScAlN [22-24]"

3. Pg. 3, Col 1: "...precision required for such steps for below-6 GHz spectra..."

A reference to support the Angstrom-precision required would be very helpful. This also necessitates a discussion on how CLMR devices with LFE are more robust to the fabrication variations than FBARs. What is the precision required for 5G/6G? What does that translate to in IDT pitch, electrode width, electrode thickness, piezo film thickness, etc.? This analysis should be added to the manuscript from both fundamental mode shape (simulation) model as well as from intra-chip measurements on identical devices and chip-to-chip variations.

Even though we are confident of the truthfulness of our claim on the level of precision required to shift the resonance of commercial FBARs, we could not find any publicly available reference supporting it. Therefore, we decided to remove it. In fact, we believe that the simple fact that FBARs do not come with lithographic definition of the resonance frequency is a considerable disadvantage compared to the proposed technology.

We do not claim that our CLMRs are more robust to fabrication variations than FBARs. Our point is that, given the intrinsic lithographic definition of the resonance frequency, CLMRs provide an extra degree of freedom compared to FBARs in allowing for a simpler and cheaper fabrication process, involving fewer

lithographic steps. Even without the lithographic definition of the resonance, our devices, being top electrode-only, allow for simpler micro-machining.

In terms of chip-to-chip and intra-chip variations studies, we are limited by the available fabrication facilities, not allowing the full fabrication on 8" substrates. In particular, we lack lithographic and etching capabilities on such substrates. This, combined with the fact that we employ a low throughput e-beam lithography tool, limits the amounts of samples we have available, and therefore a deep study on the chip-to-chip variations. Given the fact that we process chip-by-chip, we have a much larger variability compared to a process that is performed on the same wafer. We still believe that the variability we achieved is not very high, especially given the abovementioned premises, and also because the most important step, i.e., the piezo layer sputtering, is done at the wafer level. In order to reply effectively to this comment, we decided to add error bars on the plots of Fig.8, in order to show, to some extent, the variability we have in our process. The error bars take into account devices fabricated in two different batches. We modified the content of the "FEM simulations" section in Methods to describe the new figure. The change we made is described in details in the reply to comment#15 of the same reviewer.

We ask the reviewer to consider our work only as a first step towards the commercialization of our technology, since it is able to address outstanding problems that the current state-of-the-art has, but still needs a considerable amount of work before that will happen. With this in mind, we limited our claims throughout the manuscript and we addressed the next comments highlighting the limitation of our technology, for example stating that we need further design optimization to suppress the resonator spurious modes, decrease the device variability, reduce the TCF, and that the power handling still need to be improved to reach the commercial level.

In terms of FEM simulations, we decided not to add a comprehensive analysis on parameters variation, as that would require a considerable amount of time and go outside the scope of the work, focused on presenting a new technology and solving current technological outstanding problems.

4. The authors discuss the non-linear dependence of k_2 on doping concentration, stating on Pg. 4, Col 1 "Because of the non-linear change.. the maximum electromechanical coupling enhancement is achieved for high doping levels." Later in the Results section, reference to cubic/quadratic dependence determined through DFT is mentioned and referenced. However, the strict correlation to k_2 with nonlinear dependence on doping concentration isn't clear. Can the authors elaborate on the physical mechanisms (stress, etc.) behind this? It would also be good to refer to Methods Fig. 9 in that discussion.

Thanks for the comment. We believe this kind of explanation will make it clearer to a broader audience as per why the scandium doping process is responsible for our performance enhancement. We clarified on the relationship between k_2 and material parameters, such as piezo coefficients, dielectric constant, and equivalent stiffness, and added references supporting that.

We added the following sentence in the Results section:

"Larger electromechanical couplings are obtained in resonators due to the combined increase of the piezoelectric coefficients and the softening of the equivalent stiffness constant [7]."

Moreover, in the methods section we explained difference in the physical reasons under the increase in k_t^2 due to the increase in Sc-doping and due to the metal. The sentence we added is:

“In the first case, the k_t^2 increases because of an increase in the piezo coefficients and of a decrease of equivalent stiffness, according to the following equation [7]:

$$k_t^2 \propto \frac{e^2}{\varepsilon \cdot C}$$

where e is the piezoelectric coefficient, ε the dielectric constant and C the equivalent stiffness. The electromechanical coupling enhancement deriving from the utilization of different metals comes from the increased mismatch in acoustic impedance between the piezo and the electrode, resulting in a better confinement of the resonant mode inside the resonator cavity [44]”.

5. In Results: Highly-doped ScAlN section, it would be good to add comments on 8” and larger substrate deposition capabilities, uniformity, CMOS-compatibility, etc. Also in that same section, Fig. 2 needs some improvement. Specifically, in Fig. 2c it is very difficult to get anything useful out of the image. The AOG does not resolve well, particularly in print. It would be better to change this to an AFM scan with color map to highlight the defect density the authors are trying to demonstrate. AFM measurements should also replace Fig. 7 in Methods. Fig. 2d similarly does not show much. A TEM scan rather than SEM scan would be much more informative and provide better contrast. Ideally, TEM with EELS to show composition should be performed.

Thank you for this comment, we found this extremely useful in improving the quality of our publication. We clarified that deposition on 200 mm wafers is the current MEMS foundry standard, at least for RF resonators. Deposition of ScAlN on larger substrates is not possible since, at the moment, no tools with such capabilities exist. We did not add this detail in the text because we think it would go out of the scope of the publication. The comment on CMOS compatibility was added in the introduction as a reply to the reviewer’s comment number 2. Moreover, we did comment on the thickness uniformity across the wafer, and we added a thickness map of our wafer in Fig.2 that shows it. Finally, we changed the SEM top surface with two AFM scans, one at the center and one at an edge of the wafer while reporting the surface roughness, and we changed the SEM cross-section into a TEM one. Our tool does not have EELS measurement capabilities.

We added the following text in the results section (the non-bold part was already present):

The deposition was carried out with reactive magnetron sputtering directly on 200 mm high-resistivity Si<100> wafers, those being the current MEMS foundry standard wafer diameters. The target thickness was 280 nm, and a good level of thickness uniformity was obtained throughout the whole wafers, as shown in Fig.2b.

[...]

Fig.2d-e shows two Atomic Force Microscope (AFM) images of the top surface of the film. In particular, Fig.2d depicts the center of the wafer, while Fig.2e the edge, in order to show that the film has a smooth

surface and low AOG density throughout the whole substrate. The sub-nm level of surface roughness is reported in the figure caption. Finally, in Fig.2f a Transmission Electron Microscope (TEM) image of the film's cross-section is reported, showing the material stack and good degree of c-axis orientation, especially considering the low film thickness.

The updated figure 2 is:

Fig. 2 a) Industrial-grade Evatec® Clusterline II sputtering tool employed to deposit the ScAlN thin films of this work. In b), a thickness map of the deposited film is shown. The black dots are the measurement points, which were fitted with a 3D spline. The measurements were performed via ellipsometry. In c), the X-Ray Diffractometry (XRD) rocking curve scan of the Sc_{0.3}Al_{0.7}N 280 nm film with a measured FWHM value of 2.1° is reported. d)-e) show Atomic Force Microscope (AFM) top views of the film, demonstrating a low density of AOGs and sub-nm surface roughness of 0.65 and 0.61 nm, respectively. d) is a scan of the wafer center, while e) of an edge. In f), a Transmission Electron Microscope (TEM) cross-sectional view of the same film is reported, showing high c-axis orientation.

6. In Fig. 2b and Fig. 7, how does the FWHM of the authors' films compare with that of standard ion-assisted sputtered AlN, epitaxial AlN, epitaxial ScAlN, etc.? A frame of reference for the measured

FWHM would be very useful.

The authors agree that a literature comparison between sputtered AlN and ScAlN could be helpful to better frame the quality of our work. Nevertheless, it must be considered that deposition substrate, film thickness, and deposition techniques have a deep impact on film's quality. With respect to the substrate, having a crystal lattice that matches the nitride structure is of extreme help to ensure good crystallinity and piezoelectric performance. For this reason, Pt and Mo are commonly used as a bottom electrode and seed layer. However, the resonator topology described in this work requires an electrically open surface on the bottom side of the plate, to ensure optimal coupling of stress and electric field closer to the IDTs. Other works employ an AlN seed layer, that instead is not present in the current work for a matter of simplicity of the process and deposition tool constraints. Film thickness has also a strong impact on the crystallinity and the rocking curve, since thick films showcase better crystallinity. Furthermore, the Sc-doping level has also an impact on the crystallinity, being the latter in general worse for higher doping levels. Lastly, epitaxial ScAlN in general offers better rocking curves, but also suffers from larger stresses and increased fabrication costs with a limited throughput. Since our technology mandates deposition of almost stress-free thin films on Si substrates, we believe that a fair comparison can only be done between our results and AlN and ScAlN deposited according to the same criteria. Nevertheless, no data is present in the literature on the FWHM of films grown with the same sputtering process, similar thickness and doping level and absence of bottom electrode or AlN seed layer.

Nonetheless, as suggested, we tried to contextualize our result. We stated that the FWHM is, to our knowledge, the best ever achieved for a film sputter deposited, with the same doping level, similar thickness, and directly deposited on silicon, and has a value comparable to AlN films with similar thickness deposited by the authors. We made sure to include that epi-films are inevitably better in terms of crystallinity, but we chose sputtering over epitaxy since we are aiming for something that can be deposited with high throughput and cost-effectively. Finally, we cited a work reporting resonators fabricated on epi-films, and we state that our devices still achieve notably higher figures of merits thanks to our optimized fabrication process and device design.

The added sentence:

“The obtained FWHM is, to the authors' knowledge, the lowest ever achieved on a < 300 nm 30% ScAlN film sputter deposited directly on silicon without a bottom electrode or an AlN seed layer. Moreover, it shows only minor degradation compared to the ones of AlN films with similar thickness deposited by the same authors. Compared to AlN and ScAlN epitaxial films reported in the literature [42], the crystallinity is inevitably degraded. Nevertheless, the process of sputtering allows to achieve lower film stress values, higher throughput and lower manufacturing cost and therefore is desirable for mass production. Moreover, as reported in Fig.5b-e, resonators fabricated with the present sputtered film notably exceed the performance of similar devices built on epi-films, as a result of the fabrication and device level optimization performed by the authors.”

7. One known challenge for ScAlN, particularly in the case of heavily doped films such as that in this work, is scaling film thickness down while maintaining high k_2 . The k_2 dependence on h/λ (e.g.

Fig. 8a) tells us that scaling to 20 GHz will still require further thinning of the film. Discussion on this and on any limits/challenges/state of the art in ultra-thin-film ScAlN should be included in the paper.

Thanks for the comment. We understand the reviewer's concern, and therefore we decided to add in the manuscript the scaling of our devices to the higher end of the 7-20 GHz spectrum. We sputtered a thinner film (150 nm) with the same recipe of the 280 nm, achieving a 2.7 degrees FWHM. We then designed and fabricated devices in the 14-20 GHz range, showcasing minor degradation in terms of Q and k_t^2 compared to the 7-11 GHz ones, that is explainable in the lower crystallinity of the piezoelectric layer, as we demonstrated in previous publications (that are referenced in the added text). We added a new subsection in Methods, where we discuss the challenges associated with a further scaling, we show the admittance curves of the devices in a new figure and report their resonance, wavelength, 3dB Q, k_t^2 , CO, and Q_{fit} in a new table, to be consistent with the table for the 7-11 GHz devices. We state that further optimization is required. Finally, we clarify that the achieved in-plane dimensions are compatible with current state-of-the-art deep or extreme ultraviolet tools. The measured data of the resonators have also been added to the online folder for any interested reader.

The Methods subsection that was added is:

“Scaling to the higher end of the spectrum.

The Results section showcased resonators and filters up to 11 GHz of operation. Nevertheless, since the final goal is to cover the entire 7-20 GHz spectrum, further reduction of the film thickness and in-plane dimensions is required. Such scaling poses several challenges, specifically on the piezoelectric layer growth side and on the increased lithographic resolution. In order to demonstrate the capabilities of the current technology platform, a 150 nm film was deposited by employing the same sputtering recipe, yielding a 2.7° FWHM. Moreover, LFE CLMRs in the 14-20 GHz range were fabricated on it (Fig. 12) employing the same fabrication process previously described, with the only difference being an IDT thickness of 43 nm, as that value maximizes the coupling for the CLMR with the optimal h/λ ratio with the used material stack. The devices showcase Q and k_t^2 values with only minor degradation compared with the 6-11 GHz ones. In fact, as shown in Table 4, electromechanical couplings in excess of 6% in the 18-20 GHz range were measured, as well as Q_{3dB} as high as 421 at 14 GHz, and above 200 up to 18 GHz. The BVD fittings are provided. The performance degradation is explained by the lower quality of the piezoelectric layer, already seen by the authors to greatly impact both the Q and the k_t^2 of resonators [45]. In [43] in fact, the same authors show $Q_{3dB} \cdot k_t^2$ figures of merit in the 7-11 GHz range that are comparable to the ones of the 14-20 GHz presented in this section, while employing a piezoelectric layer with similar crystallinity. In [45], simply utilizing a more crystalline substrate, the figures of merit are tripled. Therefore, further piezo sputtering characterization must be made to bring the thinner film on par with the 280 nm one. Nevertheless, the present results highlight the capabilities of the present technology to cover also the higher end of the 7-20 GHz spectrum, and that the reduced in-plane dimensions and layer thicknesses do not compromise the resonator performance. In terms of mass-production of such devices, the minimum feature size achieved in this work (75 nm) is manufacturable by state-of-the-art extreme ultraviolet stepper tools.

The new figure added:

Fig. 12 Measured admittance vs. frequency for showcase resonators operating in the 14-20 GHz range, along with their BVD fittings, demonstrating the scaling capabilities of the LFE CLMR technology.

The new table added:

f_s [GHz]	λ [nm]	Q_{3dB}	k_t^2 [%]	C_0 [fF]	Q_{fit}
14.15	577	421	2.86	34	325
15.83	500	254	4.25	52	229
17.76	417	208	5.58	44	216
18.82	375	182	6.87	42	155
20	300	130	6.52	29	140

Table 4 Main metrics of the resonators shown in Fig. 12.

8. In Fig. 4, it is unclear which of these devices are standalone resonators and which are filters. It becomes more evident after Fig. 6c, but for clarity, please include labels to this effect in the figure directly. The authors should also specify in the caption the frequencies of operation for the devices shown, and comments on the bowing shown in the long (lower frequency?) resonator.

Thanks for the comment, as that will make it clearer for the readers which devices are depicted in the SEM scans. We added labeling of the different subfigures of Fig.4. We also updated the caption including the wavelength and operation frequencies of the different resonators, including an explanation for the bowing of the device in b) (residual stress-gradient, with its measured value). Finally, we stated that the device in d) is a filter and reported its center frequency.

The new caption is:

SEM pictures of fabricated resonators and filters, with highlighted relevant dimensions. a) provides an overview of different resonators on the same chip. In b), a device with a λ of 933 nm and operating at 8.35 GHz is shown. The bending is due to residual stress-gradient in the film, quantified to be 0.9 GPa/ μm . In c), a device with a λ of 560 nm and operating at 10.2 GHz is shown, while d) depicts a first-order ladder filter with center frequency at 9.17 GHz.

9. In the section on Standalone Nanoacoustic Resonators, Pg. 6, it is worth discussing the choice of metal and thickness, especially considering the nm dimensions with frequency scaling. Ultimately it is the transducer resistance and mass loading that will limit scaling. What does that look like up to 20 GHz in the LFE CLMRs?

The choice of the metal was already discussed in the Methods section, in particular in the “FEM simulations” subsection. We decided not to discuss that in Results just to keep it simple for the reader. In terms of the specific thickness choice, a sentence was added in the same subsection, as well as a reference where the same authors show how to optimize the kt2 with the right metal thickness:

“The used IDT thickness of 95 nm was chosen via FEM as the one maximizing the electromechanical coupling for the resonator with the optimal h/λ ratio of 0.4, as shown in [44].”

We believe that mass loading and transducer resistance will not limit the scaling, at least not in the 7-20 GHz range. In fact, it is true that the higher frequency resonators described in the last section have thinner IDTs (43 nm vs 95 nm), but the routing, i.e. the main contributor to the series resistance, is done in a second step with much larger thickness. Moreover, given the large number of fingers, the contribution of the IDT to the series resistance is low, given that all the fingers are electrically in parallel. We believe that the last added section (“Scaling to the higher end of the spectrum”) answers this question. In particular, the authors attribute the 14-20 GHz devices performance degradation to the lower crystallinity of the piezo layer as already reported in previous publications, and not to the increased mass loading or parasitic resistance.

10. In the same section, Pg. 6 and Fig. 5(b-e), the scatter plots of various devices indicate “similar devices all fabricated on the same chip in the same process flow.” Does this indicate spread in performance for identical designs, or rather a range of different designs with intentionally varying performance? As noted in reviewer comment #3, it is worth discussing fabrication variations and statistics for identical devices across chip and chip-to-chip.

The device metrics added to the scatter plots are for devices with slightly different designs, since the optimal ones were experimentally investigated. Nevertheless, a slight spread in performance for identical designs was noticed by the authors, and that is attributed to the fact that the devices are fabricated in an academic cleanroom facility, with a good degree of process-to-process variation that is not controllable by the authors. Therefore, further effort will be made to improve the device variability. In this work, we focus on introducing our technology, and we believe a full process variability analysis, even though interesting, would be more suitable for another publication in a more technical journal. In order to better describe the process variations, error bars were added to the plots of Fig.8, showing the current status of technology development. Moreover, we modified part of the “FEM simulations” section in Methods. A

detailed description of the changes is reported in the reply to comment #15.

11. In Fig. 5, the green highlighted regions indicate optimal performance for 6G filter synthesis. How are these performance metrics calculated or selected?

For fig.5b, the frequency range is the 7-20 GHz one. For the y-axis, we chose a minimum of 15 of $Q \cdot k_t^2$ figure of merit. The choice was semi-arbitrary, and served to set a lower bound for the performance of the devices in order to ensure an acceptable value of insertion loss in the filters. In this way, a 50ohm-matched resonator would have a motional resistance of ~ 3 Ohm. The min acceptable k_t^2 of 4% is to ensure 400 MHz of BW (as stated in the reference "Extreme massive mimo for macro cell capacity boost in 5G-advanced and 6G") at 20 GHz. In this way, the minimum acceptable Q with 4% k_t^2 is 375. Finally, the minimum $(Q \cdot k_t^2) \cdot f_s$ is $(15) \cdot 7$ GHz = 105 GHz, where 7 GHz is the minimum considered frequency (we updated Fig.5c with the new value of $Q \cdot k_t^2 \cdot f_s$ (before was 150 GHz)).

We also added a brief explanation of this in the caption of Fig.5:

The considered frequency range is 7-20 GHz, while the $Q_{3dB} \cdot k_{tSM}^2$ product of 15 is the minimum requirement to achieve acceptable values of filter insertion loss. The minimum k_t^2 of 4% is to ensure the required 400 MHz of bandwidth [] in the best case, i.e. at 20 GHz.

12. Fig. 6(d) should include an inset zoomed in on the passband. Also, what is the impedance matching for each order filter?

Thanks for raising this concern. A zoom-in of the passband was added in the figure. The matching impedances for the filters are similar to the one of F5 since the devices are synthesized starting from its building block resonators. No spurious modes are seen in the pass-band because the single-mode fitted responses of the resonators from Fig.10a-b were used instead of the measured data. We made sure to emphasize on that, also to reply to reviewer#2's comment number 4. We added in the Methods "Filter reconstruction and higher order simulation" section:

"After that, the filters are matched to impedances around 500 Ω ."

The new figure:

Fig. 6 a) Measured filter scattering parameter S_{21} vs. frequency for a bank of nanoacoustic devices fabricated on the same chip and matched via ADS® (see Table 3). In b), a comparison between unmatched (raw) data and matched one is provided, showing the same poles. In c), the pad structure is shown, together with the GS and SG probe placements to measure the filter and the series and parallel resonators. In d), ADS simulated and matched 1st, 3rd, and 5th order F5 filters are shown, starting from the measured and single-mode-fitted resonator responses (see Methods). The notch depth of the 3rd and 5th order filters is exacerbated by the employment of the simple BVD model to describe the resonator response. In a real case, the dielectric losses inherent to the piezoelectric layer [7] would cap the notch depth. Such losses were not taken into account here to be consistent with the BVD fitting used throughout this work.

13. In the section on Compact Dimensions for Handset Applications, Pg. 7, How are the number of IDT fingers and overall resonator area chosen? What is the size limit of each device based on released films (we see the bowing/curling in Fig. 4)?

The authors agree with that the reviewer that further explanation is needed in this regard. Since in this work we present a first demonstration of the capabilities of the ScAIN LFE CLMR technology, we did not match the devices to 50 Ohm, both the resonators and the filters. Therefore, in the first version of the manuscript, we wrongly reported the active area of non 50 Ohm-matched resonators. To address this comment, and to the second comment of reviewer#2, we added the following sentence:

“Being the devices matched to impedance values around 500 Ω , a 10-fold active area increase has to be considered when synthesizing filters matched to the 50 Ω standard. Nevertheless, the form factor remains contained and competitive with respect to the current state-of-the-art solutions. In addition, greater area reduction can be achieved when scaling such filters to the higher end of the 7-20 GHz range due to a reduction in the horizontal acoustic wavelength.”

We also understand the concern about the maximum achievable size due to the mechanical stability of the suspended structures. Nevertheless, we can confirm that the size does not constitute a problem. In

fact, we have design techniques in place to increase the tethered area without compromising the quality factor, but we believe an explanation of that would require a very long preamble and goes well beyond the scope of this work, which focuses on a first demonstration of the passive pass-band filters rather than a design optimization.

14. A discussion of ferroelectricity is not included in the paper. At 30% Sc doping, one would expect FE properties. Is that the case here? If so, the Methods section should also include PE loop measurements. This also prompts a discussion of power handling in the main manuscript, especially for thin films at higher frequencies.

The authors have investigated ferroelectricity on ScAlN in other publications and are quite familiar with the topic. However, we have never been able to demonstrate clear switching in lateral field configurations (such as the CLMRs reported in this work). Assuming that lateral switching is a possibility and requires the same measured coercive field of 3-5 MV/cm, and knowing the minimum gap between the IDTs at 20 GHz is around 75 nm, it would require a minimum of ~22 V to switch a device. As a comparison, 30 dBm of applied power while testing at RF correspond to ~16 V between the IDTs. Given this information, we are adamant that ferroelectricity does not play any role in the devices described in this work, since such power level will definitely cause device failure. We added a methods subsection showing the power handling capabilities as a reply to comment 1 of reviewer#2, which shows that the devices under consideration are far from handling 30 dBm of power, at least in this early stage of technology development.

15. In the Methods section on FEM Simulations, Pg. 7/8, can the authors explain the large discrepancy between DFT and measured relative permittivity from a physical/material perspective?

Thanks for the comment, we understand the concern given the large mismatch between the simulations and the experimental kt^2 values. We realized that those simulations were done with a wrong assumption: the ϵ_r of 17 that we used and reported in other works in the material science field is for films deposited on Pt, therefore not matching the one on films deposited directly on Silicon as the one of this work. Therefore, we decided to use the DFT computed ϵ_r of 14.11, closer to the one of our film as also extracted from the CO of the devices. Such value allows for a higher degree of matching between simulations and experimental results in terms of kt^2 .

The changes we made to the manuscript therefore are:

- We canceled the part referring to the extraction of the ϵ_r
- We stated that we used the ab-initio equations from Caro et al. also for the dielectric constant.
- We updated the plots of Fig.8 with the FEM simulation data using the dielectric constant from DFT ($\epsilon = 14.11$), both for the kt^2 and the resonance frequency. Moreover, we added error bars on the plots of Fig.8 to capture the variability between similar devices.

In the same section, we removed:

Fig.8 shows the agreement of measured data and simulations in terms of electromechanical coupling (a)

~~and resonance frequency (b). As it can be observed, the latter plot has an excellent agreement between measurements and simulations, especially for lower h/λ ratios. In terms of coupling, even though the trend is perfectly matched, there is a possible overestimation of the measured values since the kt^2 equation takes into account resonance and antiresonance frequencies, but the presence of some transversal spurious modes in the band shifts their values more apart from each other. Nevertheless, the single mode kt^2 value has been chosen to be consistent with the literature and to be able to make fair comparisons between devices belonging to different research works.~~

Instead, we added:

In Fig.8, a comparison between measured results and FEM simulated ones is provided in terms of electromechanical coupling (a) and resonance frequency (b) as function of the h/λ ratio. In the plot, each data point is the average between six or more devices with same h/λ fabricated in two different batches. An error bar showing the maximum and the minimum value for each geometrical ratio is also provided. As can be observed, there exists a very high agreement between FEM simulations and experimental data. In terms of k_t^2 , the measured devices showcase some variability, explained by the lack of full control of the process in this early stage of development of the technology regarding the process repeatability. Therefore, more optimization has to be performed at the fabrication level to decrease the device variability. On the other hand, the variability on the resonance frequency is low.

16. Comments/measurements on temperature dependence and stress gradients would be helpful in the Methods section.

Thanks for the comment. A new Methods subsection named “Device Temperature coefficient of frequency” was added:

The Temperature Coefficient of Frequency (TCF) of the present devices was investigated. To do so, the temperature of the chuck of the employed probe station was changed from 25 to 145°C in steps of 20°C, and the frequency shift of multiple devices was recorded. The TCF was extracted via linear fitting of the measured shifts and averaged between the different devices, resulting in -36.1 ppm/K. The obtained result shows an increase of the TCF compared to similar AlN devices [56] and that further optimization of such value needs to be performed. Possible solutions include the use of a silicon dioxide compensation layer [56].

The value of stress gradient for the used film (0.9 Gpa/um) was added in the caption of 4, to better comment on the bowing of the resonator in the b) subfigure.

17. Some typos/grammar:

a. Abstract: “...(5G FR-2) spectrum” should be changed to spectra

Corrected

b. Abstract: “yet-to-explore” should be “yet-to-be-explored”

Corrected

c. Intro: “mole of transmitted data” ?

hanged “mole” with volume

d. Pg. 2: “devices’ Radiofrequency”

Rephrased

e. Pg. 2: “it is foreseen to be refarmed in 6G...” ?

Rephrased

f. Pg. 2: “commercial produts”

Corrected

g. Pg. 2: first introduction of IDTs – reader may not know the jargon of “transducers/transduction”

Rephrased to make it clearer.

h. Pg. 2: first introduction of “the parasitics” – needs to be clarified for general reader

Rephrased to make it clearer.

i. Pg. 4: first instance of AOG in column 1 not explained until column 2.

Removed the instance with a more general sentence. In this way, the introduction remains more high-level

j. Pg. 4: “devices built with this film as active layer” should be “as the active layer”

Corrected

k. All instances of lithographic “tunability” should be changed to lithographic “definition.”

We used the term tunability to be consistent with the authors of the original paper on CLMRs, but we understand why the reviewer prefers the term “definition”. We corrected all the instances.

l. Pg. 6: “...Q3dB has been chosen as metric...”

Corrected with “was chosen”

m. Pg. 6, section on Nanoacoustic pass-band filters: Several references to Table 2 should be changed to Table 3.

Changed, there was an error in the Latex code referencing to that table

n. Pg. 7: “...In the same table, also the filters from [38]...” language grammar revision

Sentence rephrased

o. English in last sentence of conclusion

Rephrased

p. Table 3 caption: sentence ending in “one”

Rephrased

q. Pg. 9 first paragraph: sentence ending in “one”

Rephrased

r. Methods section “Devices’ Fabrication” should be changed to Device Fabrication

Corrected

s. Methods section on Bandwidth enhancement, last line “in a wider h/lambda values”

Corrected

Reviewer #2 (Remarks to the Author):

The authors report LFE CLMR's in the range of 6-11 GHz fabricated on the same chip in a 30% Sc doped ScAlN film. The results reported by the authors are indeed a giant leap forward for ScAlN Lamb wave resonators in obtaining high coupling, Q and FOM at these frequencies. In addition to some minor typos in the paper the authors need to better qualify their claims and what is still a considerable amount of research that will be required for the reported technology to be appropriate for cellular handset filters. If the authors can properly contextualize the limitations of the current technology in a revision, I would recommend publication in Nature Communications.

Thanks a lot for the encouraging words. As suggested by the reviewer, we focused our effort on contextualizing the limitations of the current state of the technology development and on making the reader aware of the amount of research effort that must be made to make the technology commercially viable. In particular, also in agreement with reviewer#1, we stressed on the required power handling improvement, the need to fix the spurious modes before increasing the filter order, the need to improve the film for covering the higher end of the 7-20 GHz spectrum, the need for a reduction in device-to-device variability, and for an improvement in the TCF.

Major comments:

1) The authors claim the reported technology is appropriate for cellular phone duplexers that isolate the transmit and receive paths. In such a case the filters must be able to handle the full transmit power on order of 0.1 to 1 W. While the small, suspended resonators reported here are unlikely to approach these values, 1 dB compression point is very quick to measure and at a minimum should be reported to properly contextualize the further work that will be needed for the technology to be applicable to cellular handsets.

Thanks for this comment, as power handling is a crucial metric for a commercial filter. We added a subsection in Methods where we show power handling tests of the filters. We did not perform only 1dB compression point measurement, as we preferred to show the device survivability instead to better highlight the limitation of the current state of our technology.

The main mechanism for the breakdown of our filters is the electromigration of the evaporated Al electrodes due to the high motional currents when high power is applied. More research work on substituting that metal with AlSiCu or a metal that is able to withstand higher powers needs to be performed. The limited value of the power handling is also explained in the matching impedance of the devices being not 50 Ohm. In fact, to match to 50 Ohm, more fingers would be designed, thus reducing the current density received by each finger and therefore increasing the power handling, as the critical current density responsible for the device breakdown would be reached at higher powers. We believe power handling is the main bottleneck of our technology. We have strategies to improve it to bring it around 18 dBm, value that is foreseen to be the requirement for mid-band 6G filters, but we would prefer not to disclose them in the current publication. Therefore, in the added Methods subsection, we explained

how the power handling tests were performed, we added a new figure showing the degradation of the matched response as function of applied power and stated that the power handling is the main bottleneck of the technology and the first issue to be addressed to bring the devices to the commercial level.

The added Methods subsection:

“Filters power handling.

The present paper poses itself as a showcase of the ScAIN LFE CLMR platform in delivering on-chip multi-frequency high-performance passband filters. Nevertheless, given the early stage of the technology development, power handling still requires a considerable amount of research work to bring it to the level required by commercial specifications. In Fig.11a, the ADS S_{21} matched scattering parameters of a measured filter are reported as function of applied input power P_{in} . The filters experience 1dB transmission characteristics degradation in the passband after the application of 9 dBm (Fig. 11b). The devices' irreversible breakdown mechanism is identified as electromigration in the aluminum IDTs due to the large alternated currents passing through the devices at their resonance (Fig.11c), especially due to the low motional resistance. Further research work is needed to improve the power handling, which constitutes the bottleneck of the current stage of the technology. Possible solutions include the employment of AlSiCu or AlSiCu-Ti-AlSiCu metal stacks instead of pure Al in order to mitigate the electromigration phenomenon, as commonly done in the CMOS foundries. Moreover, other top metals such as Pt, Mo, or Rt can be employed. Furthermore, the power handling is expected to notably improve with the use of 50-Ohm-matched filters, since the lower impedance matching requires more IDT fingers, allowing to have less current per single electrode, thus increasing the power level that generates the critical current density responsible for the devices' breakdown. “

The added figure:

Fig. 11 a) Measured S_{21} scattering parameters of a show-case filter as function of the input power. In b), the minimum IL degradation is plotted as function of the input power. The green region is the linear one, while the yellow indicates a input power region which causes a reversible IL degradation. The power levels in the red region cause irreversible device damage. In c), an SEM micrograph of the IDT fingers of the series resonator of the filter after 12 dBm were applied are shown. The device breakdown mechanism is identified as electromigration of the aluminum IDTs.

2) When the size of the filters is discussed on page 7, the reports are a bit misleading. In order to be matched to 50 ohms the area will need to be increased by 10 fold. This should be acknowledged.

Thank you for this comment. We don't want to give misleading information to a reader, and we really appreciate the possibility of correcting this. We added a sentence acknowledging this fact. Moreover, we highlighted that filters in the higher end of the spectrum would be even smaller, given the smaller acoustic wavelength. The added sentences are:

“Being the devices matched to impedance values around 500Ω , a 10-fold active area increase has to be considered when synthesizing filters matched to the 50Ω standard. Nevertheless, the form factor remains contained and competitive with respect to the current state-of-the-art solutions. In addition, greater area reduction can be achieved when scaling such filters to the higher end of the 7-20 GHz range due to a reduction in the horizontal acoustic wavelength.”

3) The spurs in the resonators and filters would not be acceptable in a cellular phone filter. The group

delay distortion would corrupt OFDM signals and would likely not be constant with input power. Can the authors acknowledge this and point to some approaches that might be used to obtain a response that is free of spurs like that seen in commercial BAW and SAW resonators/filters?

We added a note on spurious modes and how they can be detrimental to the final performance of microacoustic filters. Moreover, the concept of apodization for spurious modes suppression is introduced. The added sentences are:

“As can be observed in Fig.5, the devices showcase spurious modes between resonance and anti-resonance. Their presence is detrimental to the performance of filters, being related to unwanted ripples in the pass-band and distorting the device group delay. Such modes are identified via FEM simulations as transversal ones, well-known to originate in CLMRs [51]. As experimentally demonstrated in [51], IDT finger apodization can be employed to suppress them.”

We also explained the need for such spurious modes suppression, especially for higher order filters, in the Methods “filter simulation and performance enhancement” section, and the sentence that was added can be found in the reply to comment #4 of the same reviewer.

4) It is not clear from the paper how the simulated filter responses in Fig. 6 (d) were arrived at. The authors make it sound like they measured two resonators and then used the measurement results to construct higher order filters in ADS but I don't think that can be the case. The simulations seem to utilize spur free resonators but none of the measured resonators or 1-stage filters fit this description. Furthermore, the spurs will have a more degrading impact on filter performance for higher order filters with more stages. The authors should better describe the method used to obtain the resonator models used in the filter synthesis.

Thank you for the comment, we understand that that detail might have been misleading. The fact that the single-mode fitted responses of the devices were used instead of the raw data was actually stated in the Methods section, but we understand we should have made it clearer also in the Results section. Therefore, we added an explanation for it, clarifying that the simulated filters employ single-mode fittings of the measured devices composing the filter F5 instead of the raw data. That was done for the sake of clarity, since the spurious modes become more dominant in the filter response when higher orders are considered. We added the following sentence in the “filter simulation and performance enhancement” section:

“The S12 responses of the filters, unlike the ones of Fig. 6a, are spurious-free. The reason lies in the utilization of the single-mode fittings of the building blocks resonators instead of the measured data. In fact, the impact of spurious modes becomes more evident when the filter order is increased, and design techniques such as IDT finger apodization must be used to suppress them before synthesizing higher order devices. “

We also slightly changed the caption of Fig.6 to make it clearer. Finally, we also made it clearer in the methods section. To reply to one of the comments of reviewer#1, we added a zoom-in to the passbands of the simulated filters of Fig.6d, showing no spurious responses (that should make it also clearer to a

reader).

5) The authors tout the applicability of this technology from 7-20 GHz but show results from 7-11 GHz. It remains speculative that the frequency of this technology can be scaled to even higher frequencies while maintaining adequate performance and this should be acknowledged.

Thank you for this comment. We report the reply of comment 7 of reviewer#1, since we believe it replies to both the comments:

“Thanks for the comment. We understand the reviewer’s concern, and therefore we decided to add in the manuscript the scaling of our devices to the higher end of the 7-20 GHz spectrum. We sputtered a thinner film (150 nm) with the same recipe of the 280 nm, achieving a 2.7 degrees FWHM. We then designed and fabricated devices in the 14-20 GHz range, showcasing minor degradation in terms of Q and kt^2 compared to the 7-11 GHz ones, that is explainable in the lower crystallinity of the piezoelectric layer, as we demonstrated in previous publications (that are referenced in the added text). We added a new subsection in Methods, where we discuss the challenges associated with a further scaling, we show the admittance curves of the devices in a new figure and report their resonance, wavelength, 3dB Q , k_t^2 , CO , and Q_{fit} in a new table, to be consistent with the table for the 7-11 GHz devices. We state that further optimization is required. Finally, we clarify that the achieved in-plane dimensions are compatible with current state-of-the-art deep or extreme ultraviolet tools. The measured data of the resonators have also been added to the online folder for any interested reader.

The Methods subsection that was added is:

“Scaling to the higher end of the spectrum.

The Results section showcased resonators and filters up to 11 GHz of operation. Nevertheless, since the final goal is to cover the entire 7-20 GHz spectrum, further reduction of the film thickness and in-plane dimensions is required. Such scaling poses several challenges, specifically on the piezoelectric layer growth side and on the increased lithographic resolution. In order to demonstrate the capabilities of the current technology platform, a 150 nm film was deposited by employing the same sputtering recipe, yielding a 2.7° FWHM. Moreover, LFE CLMRs in the 14-20 GHz range were fabricated on it (Fig. 12) employing the same fabrication process previously described, with the only difference being an IDT thickness of 43 nm, as that value maximizes the coupling for the CLMR with the optimal h/λ ratio with the used material stack. The devices showcase Q and k_t^2 values with only minor degradation compared with the 6-11 GHz ones. In fact, as shown in Table 4, electromechanical couplings in excess of 6% in the 18-20 GHz range were measured, as well as Q_{3dB} as high as 421 at 14 GHz, and above 200 up to 18 GHz. The BVD fittings are provided. The performance degradation is explained by the lower quality of the piezoelectric layer, already seen by the authors to greatly impact both the Q and the k_t^2 of resonators [45]. In [43] in fact, the same authors show $Q_{3dB} \cdot k_t^2$ figures of merit in the 7-11 GHz range that are comparable to the ones of the 14-20 GHz presented in this section, while employing a piezoelectric layer with similar crystallinity. In [45], simply utilizing a more crystalline substrate, the figures of merit are tripled. Therefore, further piezo sputtering characterization must be made to bring the thinner film on par with the 280 nm one. Nevertheless, the present results highlight the capabilities of the present technology

to cover also the higher end of the 7-20 GHz spectrum, and that the reduced in-plane dimensions and layer thicknesses do not compromise the resonator performance. In terms of mass-production of such devices, the minimum feature size achieved in this work (75 nm) is manufacturable by state-of-the-art extreme ultraviolet stepper tools.

The new figure added:

Fig. 12 Measured admittance vs. frequency for showcase resonators operating in the 14-20 GHz range, along with their BVD fittings, demonstrating the scaling capabilities of the LFE CLMR technology.

The new table added:

f_s [GHz]	λ [nm]	Q_{3dB}	k_t^2 [%]	C_0 [fF]	Q_{fit}
14.15	577	421	2.86	34	325
15.83	500	254	4.25	52	229
17.76	417	208	5.58	44	216
18.82	375	182	6.87	42	155
20	300	130	6.52	29	140

Table 4 Main metrics of the resonators shown in Fig. 12.

'''

Minor comments:

1) In Fig. 7 I believe, based on the text narrative, that the left most image uses a 30 sccm N2 gas flow. The figure 7 caption, however, shows the best film (left most) was achieved with a 60 sccm gas flow. The authors should verify and correct if needed.

Thanks for spotting this. Corrected.

2) The final paragraph in the methods makes reference to a Fig. ??a.

Apologies for that. That part had to be removed from the manuscript and shouldn't have been there in the version you read. We removed it completely.

3) The notch depths in Fig. 6 d don't seem realistic. Such performance is not achieved for lower frequency 5-stage ladder filters using resonators with much larger FOM. Can the authors comment on this? Could something else limit the notch depth in a practical filter?

Thank you for this comment. We understand that it might be misleading. In order to keep the level of technicality low enough for the publication to be read by a diverse audience, instead of using a modified BVD model for the resonators, we used a simple BVD one. The latter does not include the parasitic resistance induced by the pads and the electrodes (therefore the reported Q_{fit} is a loaded one) as well as the dielectric losses in the piezo material. Such losses are described by a resistance placed either in series or in parallel to the static capacitance. Since that resistance impacts the notches of ladder filters, its lack makes the notch have an unreal depth. We did not focus on such detail in this publication, but the reviewer's concern is totally fair. In order to make it clear to any reader, we added a sentence in the caption of Fig.6:

"The notch depth of the 3rd and 5th order filters is exacerbated by the employment of the simple BVD model to describe the resonator response. In a real case, the dielectric losses inherent to the piezoelectric layer [7] would cap the notch depth. Such losses were not taken into account here to be consistent with the BVD fitting used throughout this work."

4) On page 3, the authors state "low modulus of the d31 piezoelectric coefficient". I recommend changing "modulus" to "value".

Changed.

Once more, many thanks to the reviewers for their extremely insightful comments, greatly helping the authors to convey a stronger message and increase the overall quality of this research work. We look forward to your review!

More changes that were made:

- The abstract length was reduced to comply with the max.150 words specification of Nature Journals. We word counted the abstract of several Nature Communications papers, and found that the average length is around 170 words. The current abstract has 174 words (the original one had 244), and further shrinking of the text would compromise the message we want to convey. If it will be required, though, we will further shrink it. The new abstract is:

"The last 50 years have seen an enormous growth in mobile communication, reflecting in an increasingly interconnected world. Nevertheless, demands for faster data-rates mandate a shift to higher carrier frequencies, translating to the need for more ubiquitous hardware. The 7-20 GHz range, located between the 5G sub-6 GHz and the mm-wave spectra and providing an excellent trade-off

between network capacity and coverage, constitutes a yet-to-be-explored third frequency range for 5G. Moreover, it is foreseen to become the 6G mid-band, devoted to urban area coverage. This work proposes a technological platform able to deliver CMOS-compatible, on-chip multi-frequency low-loss, wideband, and compact passband filters for cellular radios operating at mid-band frequencies, exploiting the micro-to-nano scaling of acoustic electromechanical resonators and filters. The presented results showcase the first-ever demonstrated low insertion loss bank of 7 nanoacoustic passband filters in the X-band. Most of the filters showcase fractional bandwidths above 3% and sub-dB loss per stage in an extremely compact form factor, enabling the manufacturing of X-band filters and duplexers that can be integrated in mobile handsets.”

REVIEWER COMMENTS

Reviewer #1 (Remarks to the Author):

The authors have addressed the reviewers concerns adequately and made corresponding modifications to the manuscript such that it is at the quality of publication in Nature Communication.

Reviewer #2 (Remarks to the Author):

The authors have done a thorough effort to address the reviewer feedback and I recommend publication after the minor revisions below.

1) Fig. 5b-e do not properly capture the SOA in LN A-mode, ScAlN BAW, and AlN BAW resonators operating between 6-60 GHz. I recommend the following publications be included in the graphs to provide a much more accurate representation of the SOA. While I realize that some of these papers were published after the submission of this original manuscript, the point of these charts is to reflect the state-of-the-art and doing so does not detract from the accomplishments of this paper which are distinct from these works.

a. J. Kramer et al., "57 GHz Acoustic Resonator with k^2 of 7.3 % and Q of 56 in Thin-Film Lithium Niobate," 2022 International Electron Devices Meeting (IEDM), San Francisco, CA, USA, 2022, pp. 16.4.1-16.4.4, doi: 10.1109/IEDM45625.2022.10019391.

b. <https://doi.org/10.48550/arXiv.2307.05742>

c. S. Nam, W. Peng, P. Wang, D. Wang, Z. Mi and A. Mortazawi, "A mm-Wave Trilayer AlN/ScAlN/AlN Higher Order Mode FBAR," in IEEE Microwave and Wireless Technology Letters, vol. 33, no. 6, pp. 803-806, June 2023, doi: 10.1109/LMWT.2023.3271865.

d. Izhar et al., "A K-Band Bulk Acoustic Wave Resonator Using Periodically Poled Al_{0.72}Sc_{0.28}N," in IEEE Electron Device Letters, vol. 44, no. 7, pp. 1196-1199, July 2023, doi: 10.1109/LED.2023.3282170.

e. R. Vetry et al., "A Manufacturable AlScN Periodically Polarized Piezoelectric Film Bulk Acoustic Wave Resonator (AlScN P3F BAW) Operating in Overtone Mode at X and Ku Band," 2023 IEEE/MTT-S International Microwave Symposium - IMS 2023, San Diego, CA, USA, 2023, pp. 891-894, doi: 10.1109/IMS37964.2023.10188141.

f. D. Mo, S. Dabas, S. Rassay, and R. Tabrizian, "Complementary-Switchable Dual-Mode SHF Scandium Aluminum Nitride BAW Resonator," IEEE Trans. Electron Devices, vol. 69, no. 8, pp. 4624-4631, Aug. 2022, doi: 10.1109/TED.2022.3183963.

g. W. Zhao et al., "15-GHz Epitaxial AlN FBARs on SiC Substrates," in IEEE Electron Device Letters, vol. 44, no. 6, pp. 903-906, June 2023, doi: 10.1109/LED.2023.3268863.

2) In addition, for fig 5b-e, the use of Q_s 3dB as the metric will especially benefit the resonators reported in this work, where the impedance is too large to build filters directly matched to 50 Ω . Other works with lower impedance resonators that are more suitable for realistic filters will have a lower Q_s due to the electrical resistance of the metal electrodes being larger with respect to motional impedance. A more common metric used in this field would be Q_{max} or Q_p to better decouple the reported resonator Q from the impedance value of the resonator. This should at least be mentioned in the text, perhaps with a reference, to provide context to Nature Comms readers who may not be as experienced with this topic.

3) It would be beneficial to include a reference to IDT apodization techniques on page 8 and how those are used to mitigate spurs.

4) Figure 12 appears to have a strange box around it that should be corrected.

5) The authors might consider including their data from Table 4 in Fig. 5 b-e.

September 25th, 2023
Boston MA, USA

The authors would like to further praise the reviewers' efforts to further improve the quality of the present work. All the remarks were truly appreciated, and any lingering question was addressed to the best of our knowledge.

Sincerely,
The authors

Reviewer #1 (Remarks to the Author):

The authors have addressed the reviewers concerns adequately and made corresponding modifications to the manuscript such that it is at the quality of publication in Nature Communication.

The authors would like to thank the reviewer once more for their work and evaluation of the present work.

Reviewer #2 (Remarks to the Author):

The authors have done a thorough effort to address the reviewer feedback and I recommend publication after the minor revisions below.

The authors would like to further thank the reviewer for their effort and suggestions in this second iteration. We unanimously agree that the concerns raised in the previous review and addressed in this version of the paper have helped further improving the quality of the proposed work.

Q1. Fig. 5b-e do not properly capture the SOA in LN A-mode, ScAlN BAW, and AlN BAW resonators operating between 6-60 GHz. I recommend the following publications be included in the graphs to provide a much more accurate representation of the SOA. While I realize that some of these papers were published after the submission of this original manuscript, the point of these charts is to reflect the state-of-the-art and doing so does not detract from the accomplishments of this paper which are distinct from these works.

a. J. Kramer et al., "57 GHz Acoustic Resonator with k^2 of 7.3 % and Q of 56 in Thin-Film

Lithium Niobate," 2022 International Electron Devices Meeting (IEDM), San Francisco, CA, USA, 2022, pp. 16.4.1-16.4.4, doi: 10.1109/IEDM45625.2022.10019391.

b. <https://doi.org/10.48550/arXiv.2307.05742>

c. S. Nam, W. Peng, P. Wang, D. Wang, Z. Mi and A. Mortazawi, "A mm-Wave Trilayer AlN/ScAlN/AlN Higher Order Mode FBAR," in IEEE Microwave and Wireless Technology Letters, vol. 33, no. 6, pp. 803-806, June 2023, doi: 10.1109/LMWT.2023.3271865.

d. Izhar et al., "A K-Band Bulk Acoustic Wave Resonator Using Periodically Poled Al_{0.72}Sc_{0.28}N," in IEEE Electron Device Letters, vol. 44, no. 7, pp. 1196-1199, July 2023, doi: 10.1109/LED.2023.3282170.

e. R. Vetry et al., "A Manufacturable AlScN Periodically Polarized Piezoelectric Film Bulk Acoustic Wave Resonator (AlScN P3F BAW) Operating in Overtone Mode at X and Ku Band," 2023 IEEE/MTT-S International Microwave Symposium - IMS 2023, San Diego, CA, USA, 2023, pp. 891-894, doi: 10.1109/IMS37964.2023.10188141.

f. D. Mo, S. Dabas, S. Rassay, and R. Tabrizian, "Complementary-Switchable Dual-Mode SHF Scandium Aluminum Nitride BAW Resonator," IEEE Trans. Electron Devices, vol. 69, no. 8, pp. 4624-4631, Aug. 2022, doi: 10.1109/TED.2022.3183963.

g. W. Zhao et al., "15-GHz Epitaxial AlN FBARs on SiC Substrates," in IEEE Electron Device Letters, vol. 44, no. 6, pp. 903-906, June 2023, doi: 10.1109/LED.2023.3268863

R1. Thank you for the comment and the precious suggestions provided to better identify the current state-of-the-art of acoustic resonators. We highly appreciated the chance of updating our scatter plots and, thus, we added all the devices reported in the recommended publications. Due to the recent demonstration of a novel high Figure-of-Merit device (Kramer et al.), we decided to adopt a log scale on the Y axis to better represent the SOA dispersion. Moreover, we decided to modify the quantity reported in Fig5e from Qkt^2 vs kt^2 to Q vs kt^2 plot, since it better conveys the idea that devices suitable for filter manufacturing must showcase high kt^2 (to enable a specific fractional bandwidth) and high Q (to ensure steep roll-off and high selectivity and out-of-band rejection). The authors agree with the reviewer that the figure of merit achieved by other groups do not detract from the accomplishment of this paper, as each resonator technology showcases performance and limitations that must be considered holistically when approaching the design of the next generation of RF front ends. To further highlight this point, we introduced the following sentence in the text:

"Furthermore, recently developed periodically poled films (P3F) only ensure kt^2 maximization at a single frequency, due to the strong relationship between the desired mode of vibration and the stacking necessary to sustain it. This peculiarity prevents the

manufacturing of devices with high performance capable of spanning the whole frequency band of interest.”

Q2. In addition, for fig 5b-e, the use of Q_s 3dB as the metric will especially benefit the resonators reported in this work, where the impedance is too large to build filters directly matched to 50 Ω . Other works with lower impedance resonators that are more suitable for realistic filters will have a lower Q_s due to the electrical resistance of the metal electrodes being larger with respect to motional impedance. A more common metric used in this field would be Q_{max} or Q_p to better decouple the reported resonator Q from the impedance value of the resonator. This should at least be mentioned in the text, perhaps with a reference, to provide context to Nature Comms readers who may not be as experienced with this topic.

Thank you for the insightful remarks. We are aware that in the MEMS resonator community there is no common agreement on the metrics to properly extract the quality factor. For example, many publications report Q_{3dB} even if spurious modes are close to the peak, and Q_p extraction in presence of noisy anti-resonance peaks.

Unfortunately, even the recommended publications often do not clearly state the metric adopted for the extraction of the quality factor, nor provide data that could be used to assess the devices' performance with a common definition. Therefore, we decided to add the Q_{max} in the scatterplot, alongside Q_{3dB} , for the devices presented in this work, and the maximum reported Q for the referenced devices, to try to level the ground for a fair comparison. Thus, Fig.5 caption was changed to:

“Due to the lack of a generalized guideline for quality factor extraction, the maximum Q retrieved from the referenced publications is reported. More in detail, motional Q is used for [27, 30, 32, 33, 37, 38, 51, 52], Q_{3dB} for [35, 43], Q_{max} for [53, 54], and Q_p for [55].”

Furthermore, we added a paragraph to further motivate this choice:

“The adoption of different metrics to evaluate the quality factor of a resonator (e.g. Q_{3dB} , Q_p or anti-resonance Q , Q_m or motional Q [7], and Q_{max} or Bode Q [56]), some of which require significant data manipulation or de-embedding, and the lack of a standardized criteria for its extraction significantly complicates any state-of-the-art comparison. To address this challenge, the authors decided to report the largest metric presented in each

of the the referenced publications. A clarification on which metric was used for each of them is reported in the caption of Fig.5.

According to the data reported in Fig.5, the devices presented in this work possess the highest Q3dB in the 6-20 GHz range among the cited publications. Nevertheless, the Q3dB is, by nature, a loaded one. Therefore, to make a fair comparison, we decided to add the Qmax metric. The values of Qmax are also reported in Table~II.."

Q3. It would be beneficial to include a reference to IDT apodization techniques on page 8 and how those are used to mitigate spurs.

R3. The authors agree on the importance of reporting spurious mitigation techniques to further consolidate this work. To address this comment, we added a more detailed explanation on the origin of the spurious modes (transversal modes) and on proposed apodization techniques, referencing previous works that addressed this problem on different Lamb wave resonator platforms, including AlN and Lithium Niobate. In the main text (page 6) we added:

"IDT finger apodization techniques, first introduced in SAW resonators [8] in order to control and shape the distribution of acoustic energy in the resonator's active area, were experimentally demonstrated to successfully suppress spurious resonances in CLMRs [57] and other kinds of Lamb wave resonators [58-59]."

Q4. Figure 12 appears to have a strange box around it that should be corrected.

R4. Thanks. We have fixed the figure.

Q5. The authors might consider including their data from Table 4 in Fig. 5 b-e.

R5. Thanks for pointing this out. We added the data from Table 4 into the scatter plots of Fig.5b-e and referenced to it in the paper body. Moreover, we added Qmax in the last column of Table 4.

Besides replying to the reviewers' comments, the authors polished the main text body to improve grammar and style.